# Elevating Visual Perception in Multimodal LLMs with Visual Embedding Distillation

Jitesh Jain[1,2]*   Zhengyuan Yang[2]   Humphrey Shi[1]†   Jianfeng Gao[2]†   Jianwei Yang[3]*†

[1]SHI Labs @ Georgia Tech    [2]Microsoft Research, Redmond    [3]Meta Superintelligence Labs

https://github.com/SHI-Labs/VisPer-LM

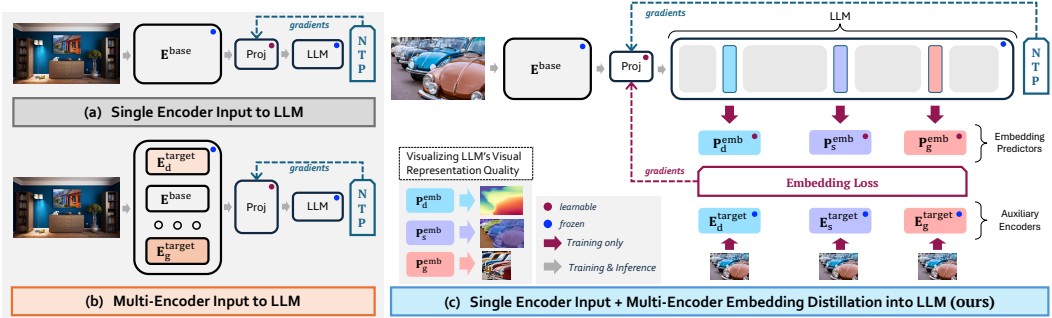

Figure 1: **Different Paradigms for Incorporating Visual Information into LLMs. (a, b)** Existing approaches [45, 67] feed features from the visual encoder(s) into the LLM and train the model solely with natural language supervision, i.e., next token prediction (NTP) to align the embedding space of the vision encoder(s) and the LLM. **(c)** We propose distilling target visual information into the intermediate representations of the LLM from a set of auxiliary vision encoders ($\mathbf{E}^{\text{target}}$). We adopt a predictive embedding [2] optimization approach at selected LLM layers during training to minimize the embedding losses and the NTP loss function, resulting in a vision-centric approach to training the Multimodal Large Language Model. We only use a single base vision encoder during inference.

## Abstract

In recent times, the standard practice for developing MLLMs is to feed features from vision encoder(s) into the LLM and train with natural language supervision. This approach often causes models to lean towards language comprehension and undermine the rich visual perception signals present in the data, which are critical for tasks involving spatial reasoning in the domain of embodied AI and robotics. Is it possible to optimize both at the same time? In this work, we propose **VisPer-LM**, the first approach that infuses visual perception knowledge from expert vision encoders into the LLM's (of an MLLM) hidden representations. We start by investigating MLLMs trained solely with natural language supervision and identify a positive correlation between the quality of visual representations within these models and their downstream performance. Given this insight, we formulate the objective during the pretraining stage in MLLMs as a coupled optimization of predictive visual embedding and next (text) token prediction. Moreover, through extensive probing, we observe improved visual representation quality due to embedding optimization, underscoring the effectiveness of our probing setup. We demonstrate that our VisPer-LM outperforms the single and multi-encoder baselines, proving our approach's superiority over explicitly feeding the corresponding features to the LLM. In particular, VisPer-LM boosts performance by an average margin of up to **2.5**% on various benchmarks, with a notable improvement of **8.7**% on the Depth task in CV-Bench.

---

*Work done during JJ, JY's time with Microsoft Research.†Equal advising.

39th Conference on Neural Information Processing Systems (NeurIPS 2025).

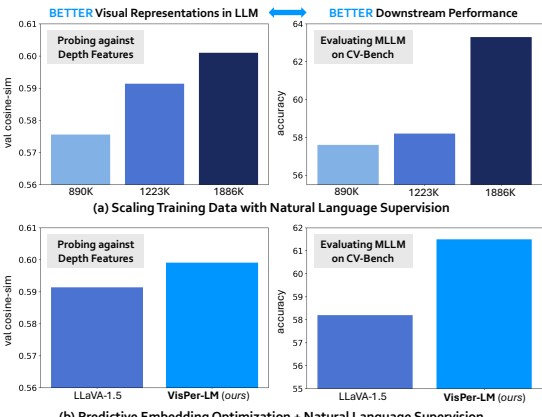

Figure 2: **Probing reveals a positive correlation between depth representation quality and performance on CV-Bench.** **(a)** Increasing training data and using only the next-token prediction objective improves visual representation quality in the LLM, as well as downstream performance. **(b)** Our method, **VisPer-LM**, outperforms LLaVA-1.5 [45] in both probing and downstream tasks under the same settings.

# 1  Introduction

In the last couple of years, Multimodal Large Language Model (MLLM) [45, 81] development has witnessed rapid growth, owing mainly to the increasing number of powerful LLMs [65, 64, 18, 66] and large-scale datasets [67, 70, 11]. A visual reasoning MLLM generally has three main components: vision encoder(s), a projector, and a decoder LLM. Given the presence of the decoder LLM responsible for producing the final outputs, the established practice to train MLLMs is with a next text token prediction objective [56] on relevant image-text datasets [37, 67] with supervision from the ground-truth text. Specifically, training an MLLM typically involves two primary stages [46]: (i) Pre-Training (PT), training a projector to align the embeddings from the vision encoder(s) to those inside the LLM, and (ii) Instruction Fine-Tuning (IFT), training the projector and LLM on conversation data for downstream instruction following tasks. Traditional MLLMs [46] usually use a pre-trained visual encoder like CLIP-ViT [55, 14, 46] to process the visual inputs.

Although effective on general visual reasoning tasks, the visual encoder's features usually lack the fine-grained visual information required to perform tasks like spatial/depth reasoning [68, 71, 32], which are important visual perception abilities. Consequently, several recent works propose simply scaling the number of visual (input) encoders [67, 41, 35, 32] to incorporate auxiliary information amicable for improving the model's visual perception abilities. However, existing works often require relatively large amounts of training data [35, 67] and compute [16, 1] for convergence, making these impractical for resource and data limited settings. Moreover, gains from using multiple visual encoders often come at the cost of latency during inference. In this work, we hypothesize that scaling the number of visual inputs and/or training data improves visual perception performance owing to its positive effect on the visual representations inside the LLM. Therefore, we explore a hidden opportunity to optimize the visual quality of representations inside the LLM directly. To that end, we propose to distill knowledge from a set of expert visual encoders into the LLM's representations during the PT stage through a set of embedding losses (Fig. 1c). During inference, we only use a single encoder, resulting in a better trade-off between visual understanding performance and efficiency as compared to feeding multiple visual inputs to the LLM (Fig. 1a,b).

We first conduct extensive experiments to establish a relationship between the (M)LLM's visual representation quality and its downstream VQA performance. To that end, we train a probe at each layer inside the LLM (of the MLLM, LLaVA-1.5 [45], in our case due to its wide usage in the academic community) to analyze their quality against expert visual features. We choose features from visual encoders trained for three tasks as targets: image segmentation [31, 36], depth estimation [74, 75], and image generation [59, 57], owing to their well-studied, fundamental nature and the first two being seminal perception abilities [27, 21]. Based on our choice of target features, we refer to benchmarks [67] evaluating the depth and spatial reasoning ability as our target tasks, i.e., we aim to improve the model at visual perception without harming its general reasoning abilities. We find that the LLM representations also improve with more data, indicating enhanced visual perception ability and downstream performance, proving the effectiveness of our probing setup (Fig. 2).

From our probing experiments, we also observe that the middle LLM layers are optimal for embedding visual information inside the LLM based on the layer-wise representation quality trend. Consequently, we investigate the effect of optimizing the intermediate LLM representations against the expert visual features at specific layers. Inspired by the recent works in embedding predictive

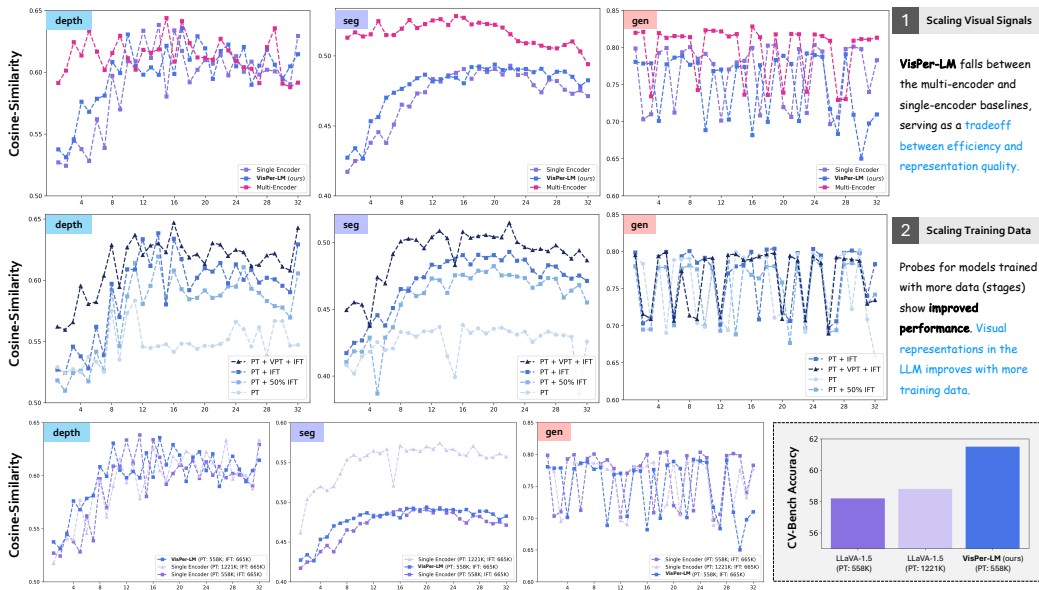

Figure 3: **Probing Visual Representation Quality across LLM layers in MLLMs**. **(1)** As shown in the first row, the multi-encoder baseline has the best probing performance owing to the additional feature inputs. The performance of probes trained on our VisPer-LM falls between the two baselines, demonstrating the effectiveness of our embedding distillation approach in learning an improved projector while only using a single encoder during inference. **(2)** We observe that the probing performance for single-encoder models trained solely with natural language supervision improves as the training data of the base MLLM increases, indicating that the LLM improves its visual representations of the world with more training data. In the last row, we observe that our VisPer-LM (base setting) outperforms LLaVA-1.5 trained with more data during the PT stage, demonstrating the effectiveness of our approach with limited (data/compute) resources.

architectures for self-supervised learning [2], we propose **VisPer-LM** (**Vis**ual **Per**ception **L**anguage **M**odel) to distill [28] knowledge from the expert visual encoders into the LLM's representations during the pre-training (PT) stage. Specifically, we optimize an embedding loss between the target feature and the embedding predictor output at the corresponding LLM layer (Fig. 1c). Note that we still use features from a base encoder as inputs to the LLM. Furthermore, we incorporate a specialized set of tokens, $\langle t \rangle$, enriched with target information into the LLM's input sequence, fostering an implicit visual chain of thought [24, 78] while enhancing the model's ability to handle spatial and depth reasoning queries. Our experiments in Sec. 5 illustrate the effectiveness of our approach on various benchmarks while outperforming the baselines. To summarize, our contributions are:

- Through probing existing MLLMs, we observe a positive correlation between visual representation quality and downstream performance on benchmarks like CV-Bench. To the best of our knowledge, we are the first to analyze the visual quality of MLLMs' representations.
- We present **VisPer-LM**, an approach to distill knowledge from expert visual encoders into LLM's representations to improve the model's visual (depth/spatial) perception abilities.
- We conducted extensive experiments to demonstrate the superiority of VisPer-LM over the corresponding single and multi-encoder baselines on various benchmarks, including boosts up to 8. 7% and 5. 6% on the depth and distance task in CV-Bench, respectively.

## 2 Related Works

### 2.1 MLLMs for Visual Reasoning

Contemporary MLLMs have three primary components: vision encoder(s), projector, and LLM. One line of work [81, 37, 45, 39, 46] uses a single pretrained vision encoder [55, 20, 50, 14, 79] and trains a projector, like an MLP [45] or QFormer [17] to align the visual features with the LLM.

A few recent approaches [63, 19] attempt to develop native MLLMs by directly feeding image patches into the LLM without using any visual encoder. Some works [1, 16] train cross-attention modules inside the LLM, modifying the LLM architecture and often requiring millions of training data samples. Another line of work explores using either multiple encoder features [67, 80, 61, 41] or multiple visual modalities [32, 47, 13] as inputs to the LLM for improved visual (spatial) reasoning performance. Our approach uses a single-base vision encoder while distilling information from expert visual encoders into the LLM's representations.

## 2.2 Perception Probing in Foundation Models

OthelloGPT [40] probed the features from a GPT-2 [56] trained on sequences from a board game, Othello, and found that the probes were able to learn the board state, indicating the ability of sequence models to learn spatial world representations. A recent work probes the features from foundational vision encoders [3] for 3D tasks. In our work, we probe the representation from the LLM layers against visual features from expert perception models. Furthermore, we establish a relationship between visual representation quality inside the LLM and downstream VQA performance.

## 2.3 Self-Supervised Learning

Distilling information [28] from a target encoder into a source encoder is a well-established technique to improve the source encoder's embeddings for a downstream task [9, 10, 25]. Recently, I-JEPA [2] proposed an embedding predictive architecture to improve representations inside a source encoder by comparing the target encoder features and mapped source encoder features with a trained predictor. A concurrent work, REPA [76] improved DiTs [54] at image generation by distilling information from DINOv2 [53]. Recently, a few works [58, 71] first distill information from target encoders into a single model and then leverage the resulting model as the vision encoder in an MLLM, requiring relatively longer and more training data for distillation. Unlike previous works, we distill information from multiple vision encoders into the LLM embedding space during the PT stage, resulting in a more vision-centric training approach without any extra training data.

## 3 Visually Probing LLM's Embeddings

In this section, we systematically analyze the quality of LLM representations within the MLLM through probing with the expert visual features as the targets. We define high-quality representations as those that map easily to the target feature space, demonstrated by a high cosine similarity to the corresponding expert encoder's features. We choose target encoders from models trained for three visual tasks: image segmentation, generation, and depth estimation, guided by the mentioned tasks' fundamental and interpretable nature, i.e., visualizing the representations' quality through the respective decoders. We use the encoder outputs from Swin-L OneFormer [31, 49], DINOv2-L Depth Anything-v2 [20, 75], and CLIP-ViT-L unCLIP-SD-2.1 [57, 55] as the segmentation ($\mathbf{E}^{\text{seg}}$), depth ($\mathbf{E}^{\text{depth}}$) and generation ($\mathbf{E}^{\text{gen}}$) as target probe features, respectively.

**Probing Setup.** We train a single-layer Perceiver Resampler [30] as the probe head at every LLM layer for each of the three target features. We use a resampler probe head (similar to the Emb Predictor shown in Fig. 4) to accommodate the different sequence lengths of the target features and representations inside the LLM. For each LLM layer, we input learnable queries into the probe head while using the layer's hidden states (for all tokens) as keys for cross-attention inside the resampler block. During probing, we set the learning rate as $1e^{-3}$ and minimize the smooth-L1-loss objective with a batch size of 256. We train the probes for two epochs on the 118k images from the COCO-train2017 [43] set with the text query: *"Describe the image in two lines."*. We find a positive correlation of 0.98 between the depth probing performance and CV-Bench accuracy in Tab. X, proving the effectiveness of our probing setup. We analyze the following LLaVA-1.5-based models with Llama-3-8b [64] as the LLM and CLIP-ConvNeXT-XXL [14] as the visual encoder:

- Single-Encoder MLLM with features from $\mathbf{E}^{\text{base}}$ passed through an MLP into the LLM.
- Multi-Encoder MLLM with features from $\mathbf{E}^{\text{base}}$, $\mathbf{E}^{\text{gen}}$, $\mathbf{E}^{\text{depth}}$, and $\mathbf{E}^{\text{seg}}$ concatenated in the feature dimension (in order) and passed through an MLP into the LLM.
- **VisPer-LM** with features from $\mathbf{E}^{\text{base}}$ passed through an MLP into the LLM and information from $\mathbf{E}^{\text{gen}}$, $\mathbf{E}^{\text{depth}}$, and $\mathbf{E}^{\text{seg}}$ distilled into the LLM representations.

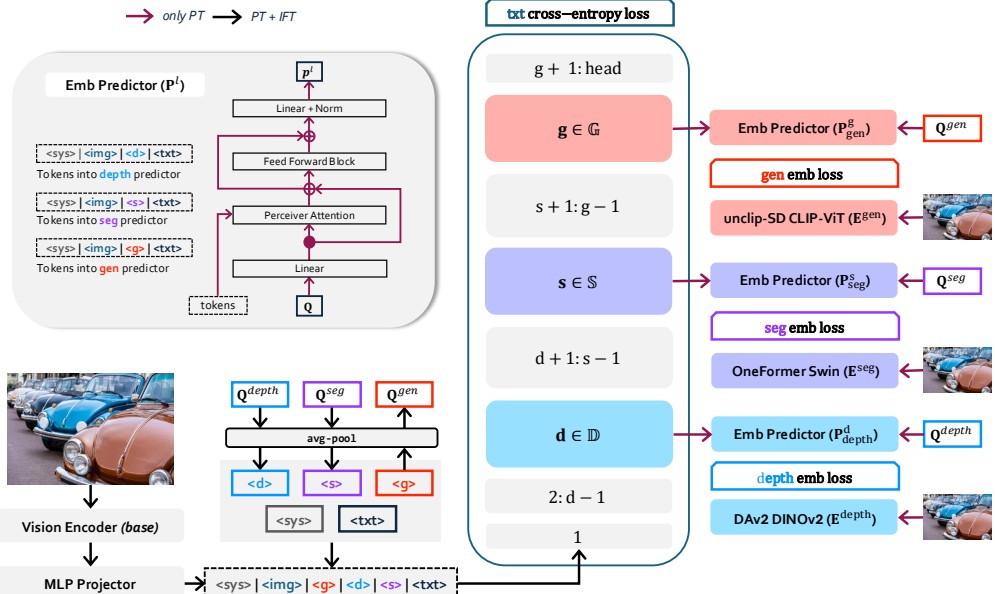

Figure 4: **Architecture for VisPer-LM**. During Pre-Training (PT), we optimize an embedding loss at specific layers for each target encoder: layers $d \in \mathbb{D}$, $s \in \mathbb{S}$, and $g \in \mathbb{G}$ for the depth, segmentation, and generation tasks, respectively. We use a resampler-based embedding predictor [30], denoted as $\mathbf{P}^l_{\{task\}}$ at each layer $l$, to output predictions. Each predictor takes in two inputs: a set of learnable queries ($\mathbf{Q}^{task}$) and the token sequence from layer $l$, with special tokens for other tasks omitted. The task tokens are derived from the corresponding embedding predictor's learnable queries. During IFT, we train with only the next-token prediction objective while keeping the special tokens frozen to not affect their nature as we found it to perform empirically better in Sec. 5.

We compute the cosine similarity between the probe outputs and the corresponding target features over the 5k images from the COCO-val2017 [43] set to get the probing performance for evaluation.

**Layerwise Trend.** Upon evaluating the probes in the single encoder baseline, we observe that the middle (12–24) layer probes show the best representation quality for the depth and seg probing tasks, with an upward trend in quality in the initial layers and a downward trend in the deeper layers, as shown in Fig. 3. We attribute the observed trend to the fact that the middle layer representations contribute the most to the (visual) reasoning (primarily spatial/depth) in LLMs [33, 34, 62]. It is an important finding when deciding the position of the embedding losses, since optimizing the middle layer representations should be optimal. Surprisingly, the probes trained to predict the features from $\mathbf{E}^{gen}$ have fairly high cosine similarity (greater than 0.7) for all layers. We attribute the mentioned phenomenon to the choice of $\mathbf{E}^{gen}$ that already has language-aligned features, unlike $\mathbf{E}^{depth}$ and $\mathbf{E}^{seg}$.

**Visual Encoding Approach.** As shown in the first row of Fig. 3, the probes for multi-encoder MLLM expectedly learn better representations than the probes for the single-encoder MLLM. Interestingly, for the former, the depth and seg probe performance remain at the same level till layer 20 and then follow a downward trend, indicating the possibility of the deeper layer representations becoming text-rich in deeper layers [44]. We do not observe a downward trend for gen probing, owing to the text-aligned target gen features. We observe that the VisPer-LM probes fall between the two baselines, serving as a good trade-off between efficiency and visual representation quality.

**Training Data.** We study the effect of scaling training data on the visual representation quality inside the single-encoder MLLM in the second row of Fig. 3. We analyze four models: (i) PT: model after the PT stage, (ii) PT + 50% IFT: model with complete PT and trained on 50% IFT data, (iii) PT + IFT: model with complete PT and IFT, and (iv) PT + VPT + IFT: model with an extra training stage on ALLaVA-Caption [6] data, during which the whole model is trained. We observe that with an increase in the amount of training data for the probed model, the probes show a gradual improvement, indicating that the representations of the visual world inside MLLMs and, therefore, downstream performance improve with just natural language supervision on more data (Fig. 2a)!

For experimental completeness, we also report probing for the single-encoder LLaVA-1.5 trained with additional ALLaVA-Caption [6] data during the PT stage. As illustrated in the third row of Fig. 3, while the inclusion of additional PT data improves LLaVA-1.5's performance on CV-Bench, our VisPer-LM achieves superior results with lesser PT data, highlighting the effectiveness of our approach with limited data. Notably, there is a significant improvement in representation quality for the seg probing task with the use of ALLaVA-Caption during PT, which we attribute to the dataset's detailed captions fostering enhanced semantic alignment [6]. Please refer to Tab. I for more results.

## 4 Embedding Visual Information into LLM

The current trend in developing MLLMs for visual reasoning is gluing vision encoder(s) to a decoder LLM with a projector and training the model to minimize the cross-entropy loss for classification over the LLM's vocabulary for next token prediction (NTP). In our attempt to add a vision perspective to training MLLMs while only using a single visual encoder during inference, we aim to optimize a predictive visual embedding objective for the intermediate LLM representations with the standard NTP objective during the PT stage. In other words, we distill auxiliary visual (task) information into the LLM's intermediate representations rather than feeding the visual features into the LLM from the expert encoders. We hypothesize that such an approach leads to a better projector initialization (Tab. VIII) for the IFT stage. Our hypothesis is validated in Fig. 3, as probes for our VisPer-LM significantly outperform the single encoder baseline, especially in the initial layers. During IFT, we train the MLLM using the NTP objective, without the embedding predictors.

### 4.1 Multi-Encoder Feature Distillation

Given a set of target visual encoders, $\mathbf{E}^{depth}$, $\mathbf{E}^{seg}$, and $\mathbf{E}^{gen}$, we distill information from their feature outputs into the LLM's representation space by minimizing embedding losses between predictor outputs and target features at certain layers. Since the token sequence inside the LLM has a different length as compared to the target representations, we use a single-layer Perceiver Resampler [30] as the embedding predictor ($\mathbf{P}^l_{\{task\}}$) that takes as inputs: learnable latent queries ($\mathbf{Q}^{\{task\}}$) along with the embeddings from layer $l$ of the LLM and outputs a prediction, $\mathbf{p}^l$. We set the number of queries in $\mathbf{Q}^{\{task\}}$ such that it matches the number of tokens in the corresponding $\mathbf{E}^{\{task\}}$ features, *i.e.*, 576, 576, and 1, for $\mathbf{Q}^{depth}$, $\mathbf{Q}^{seg}$, and $\mathbf{Q}^{gen}$, respectively.

To amplify the effect of the contextual information about the target task (spatial/depth reasoning) inside the LLM [7, 24], we use a set of special $\mathbf{N}$ task tokens, $\langle t \rangle$. We append $\langle t \rangle$ to the image tokens in the LLM's input sequence. Specifically, we average pool $\mathbf{Q}^{depth}$ and $\mathbf{Q}^{seg}$ into $\mathbf{N}$ number of tokens to obtain the $\langle d \rangle$ and $\langle s \rangle$ tokens, respectively. Since the number of target tokens for generation features is only one, we average pool $\langle g \rangle$ to obtain $\mathbf{Q}^{gen}$. During PT, we only train the MLP projector, embedding predictors, and the special tokens $\langle t \rangle$. We extract the tokens corresponding to the system prompt, input image, the corresponding $\langle t \rangle$, and the text query from a layer's output sequence as the input keys to the embedding predictor, as shown in Fig. 4.

### 4.2 Predictive Embedding Optimization

At the core of our approach is indirectly optimizing the projector during the PT stage by minimizing an embedding loss for each target representation at specific layers. As shown in Fig. 4, we feed the outputs from the $\mathbf{d} \in \mathbb{D}$, $\mathbf{s} \in \mathbb{S}$, and $\mathbf{g} \in \mathbb{G}$ layers of the LLM into the corresponding embedding predictor to obtain the predictions for computing an embedding loss ($\mathcal{L}_{emb}$), which is a weighted sum of Smooth-L1-Loss [23] and contrastive (InfoNCE) loss [69]. We denote the sets of layers for embedding distillation from $\mathbf{E}^{depth}$, $\mathbf{E}^{seg}$, and $\mathbf{E}^{gen}$ as $\mathbb{D}$, $\mathbb{S}$, and $\mathbb{G}$, respectively. The final embedding loss for each target feature is a sum of losses over all corresponding layers. We compute the final loss during PT as the sum of the NTP objective and embedding losses, as shown in Eq. (4).

We denote $\mathbf{p}^l$ and $\mathbf{t}$ as the embedding predictor ($\mathbf{P}^l_{\{task\}}$) outputs at layer $l$ and the target task features, respectively. $\mathbf{p}^l_i$ denotes the $i^{th}$ element in a batch of embeddings with a global batch size of $B$ aggregated over all GPUs. We denote $\tau$ (initialized to 2.0) as the learnable scaling factor for the contrastive loss. The weights for smooth-L1-and contrastive losses are $\lambda_{sL1} = 1$ and $\lambda_{contrastive} = 0.3$, respectively, at all selected layers. We select these values to ensure their magnitudes are comparable

Table 1: **Comparisons to Baselines.** Our VisPer-LM outperforms the single and multi-encoder LLaVA-1.5 [45] by up to 2.5% and 0.9% on average across various benchmarks, respectively. The best numbers are set in **bold** for every base-encoder and decoder LLM combination.

| | | CV-Bench | | | | General | | | |
|---|---|---|---|---|---|---|---|---|---|
| Method | Encoder | Count[2D] | Depth[3D] | Relation[2D] | Distance[3D] | MMStar | RWQA | OK-VQA | Avg |
| *Phi3-4k-mini* | | | | | | | | | |
| LLaVA-1.5 | CLIP-ViT-L | **52.4** | 67.2 | 75.2 | 56.3 | **36.5** | 57.1 | **56.7** | 57.3 |
| **VisPer-LM** (ours) | CLIP-ViT-L | **52.4** | 68.7 | 76.0 | 56.7 | 36.0 | 58.0 | 56.4 | **57.7** |
| LLaVA-1.5 | CLIP-ConvNeXT-XXL | 51.8 | 70.8 | 74.0 | 55.3 | 36.4 | 58.0 | 55.9 | 57.4 |
| **VisPer-LM** (ours) | CLIP-ConvNeXT-XXL | 49.4 | **72.5** | **77.2** | **60.3** | **38.4** | **58.4** | 56.5 | **58.9** |
| *Llama3-8b* | | | | | | | | | |
| LLaVA-1.5 | CLIP-ViT-L | 50.4 | 73.3 | 64.9 | 48.7 | 38.8 | 57.8 | 56.9 | 55.1 |
| LLaVA-1.5 (feat concat.) | CLIP-ViT-L + $\mathbf{E}^{\text{depth}}$ + $\mathbf{E}^{\text{seg}}$ + $\mathbf{E}^{\text{gen}}$ | 45.3 | 75.5 | **70.9** | 54.3 | 36.1 | 57.5 | 58.3 | 56.8 |
| LLaVA-1.5 (token concat.) | CLIP-ViT-L + $\mathbf{E}^{\text{depth}}$ + $\mathbf{E}^{\text{seg}}$ + $\mathbf{E}^{\text{gen}}$ | 45.9 | **75.7** | 68.9 | 52.7 | 37.8 | 56.5 | **59.3** | 56.7 |
| **VisPer-LM** (ours) | CLIP-ViT-L | 51.3 | 74.2 | 69.4 | **54.3** | **39.5** | 57.9 | 56.6 | **57.6** |
| Cambrian-1 [67] | CLIP-ConvNeXT-XXL + CLIP-ViT-L + SigLIP-SO-400M [79] + DINOv2-G [53] | 57.2 | 65.0 | 63.2 | 50.1 | 34.5 | 53.2 | — | — |
| LLaVA-1.5 | RADIO-ViT-L [58] | 56.4 | 64.5 | 65.8 | 50.1 | 36.6 | 55.2 | 57.4 | 55.1 |
| LLaVA-1.5 | CLIP-ConvNeXT-XXL | 54.1 | 62.8 | **69.5** | 49.8 | 37.4 | **57.5** | 56.3 | 55.3 |
| **VisPer-LM** (ours) | CLIP-ConvNeXT-XXL | **57.4** | 71.5 | 66.8 | 52.8 | 38.5 | 55.0 | **59.0** | 57.3 |

Table 2: **Results on Additional Benchmarks.** Our VisPer-LM outperforms LLaVA-1.5 on classical benchmarks like POPE [42] and GQA [29] showing its intact general reasoning ability.

| Method | Encoder | POPE | GQA | MMMU[val] | VizWiz[val] | Avg |
|---|---|---|---|---|---|---|
| LLaVA-1.5 | RADIO-ViT-L [58] | 86.3 | 62.8 | 36.9 | 50.5 | 59.1 |
| LLaVA-1.5 | CLIP-ViT-L | 85.9 | 63.5 | 38.6 | 50.6 | 59.7 |
| **VisPer-LM** (ours) | CLIP-ViT-L | **86.4** | **63.7** | **38.7** | **54.0** | **60.7** |

Table 3: **Scalability with VPT (more training data).** VisPer-LM outperforms LLaVA-1.5 on average across different CV-Bench tasks. We use CLIP-ConvNeXT-XXL as the base encoder.

| Method | LLM | Count[2D] | Depth[3D] | Relation[2D] | Distance[3D] | Overall |
|---|---|---|---|---|---|---|
| LLaVA-1.5 | Phi3-4k-mini | 49.7 | 70.0 | 72.6 | **58.7** | 61.8 |
| **VisPer-LM** (ours) | Phi3-4k-mini | **53.7** | **72.0** | **73.1** | 58.5 | **63.4** |
| LLaVA-1.5 | Llama3-8b | 56.3 | **76.8** | **73.1** | 50.3 | 63.3 |
| **VisPer-LM** (ours) | Llama3-8b | **60.0** | 75.0 | 70.8 | **55.2** | **64.6** |

and their weighted sum aligns in scale with $\mathcal{L}_{\text{NTP}}$ at convergence to maintain training stability. We set $\lambda_{\text{depth}}$, $\lambda_{\text{seg}}$, and $\lambda_{\text{gen}}$ as 0.5.

$$\mathcal{L}_{\text{sL1}}^{l}(\mathbf{p}^l, \mathbf{t}) = 0.5 \cdot (\mathbf{p}^l - \mathbf{t})^2 \text{ if } |\mathbf{p}^l - \mathbf{t}| < 1 \text{ else } |\mathbf{p}^l - \mathbf{t}| - 0.5 \tag{1}$$

$$\mathcal{L}_{\text{contrastive}}^{l} = -\log \frac{\exp(\text{sim}(\mathbf{p}_i^l, \mathbf{t}_i)/\tau)}{\sum_{j=1}^{B} \exp(\text{sim}(\mathbf{p}_i^l, \mathbf{t}_j)/\tau)} \tag{2}$$

$$\mathcal{L}_{\text{emb}}^{\mathbb{D}/\mathbb{S}/\mathbb{G}} = \sum_{l \in \mathbb{D}/\mathbb{S}/\mathbb{G}} \left( \lambda_{\text{sL1}} \mathcal{L}_{\text{sL1}}^{l} + \lambda_{\text{contrastive}} \mathcal{L}_{\text{contrastive}}^{l} \right) \tag{3}$$

$$\mathcal{L}_{\text{PT}} = \mathcal{L}_{\text{NTP}} + \lambda_{\text{depth}} \mathcal{L}_{\text{emb}}^{\mathbb{D}} + \lambda_{\text{seg}} \mathcal{L}_{\text{emb}}^{\mathbb{S}} + \lambda_{\text{gen}} \mathcal{L}_{\text{emb}}^{\mathbb{G}} \tag{4}$$

**Other Architecture Details.** As shown in Fig. 4, we use DINOv2-L [53] from Depth Anything v2 [75] as $\mathbf{E}^{\text{depth}}$, Swin-L [49] from OneFormer [31] (trained on COCO-train2017 [43]) as $\mathbf{E}^{\text{seg}}$, and CLIP-ViT-L [55] from unCLIP-SD-2.1 [57] as $\mathbf{E}^{\text{gen}}$. In the input sequence to the LLM, we append the gen, depth, and seg tokens in that order to the image tokens. Therefore, given an image-text pair, the input token arrangement is $\{\langle sys \rangle | \langle img \rangle | \langle g \rangle | \langle d \rangle | \langle s \rangle | \langle txt \rangle\}$, where $\langle sys \rangle$, $\langle img \rangle$, and $\langle txt \rangle$ denote the tokens corresponding to the system prompt, image, and text query, respectively. The target feature dimensions from $\mathbf{E}^{\text{gen}}$, $\mathbf{E}^{\text{depth}}$, and $\mathbf{E}^{\text{seg}}$ are $(1, 1024)$, $(576, 1024)$, and $(576, 1536)$, respectively, corresponding to their final layer outputs.

# 5 Experiments

In this section, we first provide a comprehensive comparison of our method's performance to that of the base MLLM, LLaVA-1.5 [45] across different base vision encoder and decoder LLM choices in Tab. 1. Next, we provide experimental results with a 2.5-stage training strategy, composed of an extra Visual Pre-Training (VPT) stage that involves training on the ALLaVA-Caption-663K [6] dataset to

Table 4: **Comparison on the Perception tasks using BLINK [22] benchmark.** Our VisPer-LM improves significantly over LLaVA-1.5 [45] on depth and relation reasoning, our target perception tasks.

| Method | Avg | Spatial Relation | Relative Depth | Count |
|---|---|---|---|---|
| LLaVA-1.5 | 55.6 | 72.0 | 51.4 | 43.3 |
| RADIO-L LLaVA-1.5 | 55.8 | 65.7 | **52.4** | **49.2** |
| Cambrian-1 | 56.9 | 74.1 | 51.6 | 45.0 |
| **VisPer-LM** (ours) | **58.8** | **75.5** | 51.6 | **49.2** |

Table 5: **Ablations on Layer sets for $\mathcal{L}_{emb}$.** Setting $\mathbb{D}=\{8, 20\}$, $\mathbb{S}=\{10, 18\}$, and $\mathbb{G}=\{12, 20\}$ performs the best. Following the findings in Sec. 3, we only experiment with middle layers.

| $\mathbb{D}$ | $\mathbb{S}$ | $\mathbb{G}$ | CV-Bench$^{2D}$ | CV-Bench$^{3D}$ | MMStar | Avg |
|---|---|---|---|---|---|---|
| {20} | {18} | {20} | 57.6 | 60.8 | 38.8 | 52.4 |
| {8;20} | {10;18} | {12;20} | **58.6** | **64.2** | 39.5 | **54.1** |
| {18;20} | {18;20} | {16;20} | 55.8 | 59.5 | **40.8** | 52.0 |
| {16;18;20} | {16;18;20} | {16;18;20} | 56.8 | 61.3 | 37.0 | 51.7 |

Table 6: **Training stages for $\mathcal{L}_{emb}$.** Using the embedding losses only during PT is optimal.

| PT | IFT | CV-Bench$^{2D}$ | CV-Bench$^{3D}$ | MMStar | RWQA | Avg |
|---|---|---|---|---|---|---|
| single-encoder | | 56.0 | 61.0 | 38.8 | 57.8 | 53.4 |
| | | 57.7 | 62.9 | 38.8 | 57.5 | 54.2 |
| ✓ | | 58.6 | **64.2** | **39.5** | **57.9** | **55.1** |
| ✓ | ✓ | **59.1** | 58.3 | 38.3 | 56.2 | 53.0 |

demonstrate the scalability of our approach to more data. Lastly, we methodically study various design factors, including the optimal choice of layers for embedding additional visual information and the number and nature of special tokens ($\langle t \rangle$) through a series of ablations.

## 5.1 Implementation Details

**Training.** During the PT stage, we use the LLaVA-558K [45] dataset to train our model for an epoch with $lr$ of $1e^{-3}$. We only train the (MLP) projector, the embedding predictors, and the special tokens ($\langle t \rangle$). During the IFT stage, we use the LLaVA-665K [45] dataset and train the projector and LLM for one epoch with an $lr$ of $2e^{-5}$ with the vision encoder and $\langle t \rangle$ kept frozen. When using VPT, we leverage the ALLaVA-Caption-663K [6] dataset to train the whole model (except $\langle t \rangle$) for one epoch with an $lr$ of $2e^{-5}$. Unless mentioned otherwise, we report results with models trained with PT and IFT stages. During the PT stage, we train our model with the NTP objective and the embedding prediction objectives. During the VPT and IFT stages, we solely use the NTP objective.

We train all our models on 16 AMD 192G-MI300X GPUs with a batch size of 256 during PT and 128 during IFT and VPT. We use CLIP-ViT-L [55] and Llama3-8b [64] as the base vision encoder and decoder LLM unless mentioned otherwise. By default, we set $\mathbf{N}$, $\mathbb{D}$, $\mathbb{S}$, and $\mathbb{G}$ to 8, $\{8, 20\}$, $\{10, 18\}$, and $\{12, 20\}$, respectively. For other hyperparameters, we follow LLaVA-1.5 [45].

**Evaluation.** We primarily evaluate VisPer-LM for vision-centric abilities on CV-Bench [67] and report results for all four tasks in CV-Bench: *Count* (2D), *Relation* (2D), *Depth* (3D), and *Distance* (3D). For ablations, we report the average accuracy over the 2D (*count* and *relation*) and 3D (*depth* and *distance*) task categories. We also evaluate our models' general visual reasoning capabilities on the MMStar [8], RWQA [72], and OK-VQA [52] benchmarks. We select MMStar as our primary benchmark for general visual reasoning, as it addresses the well-known limitations [8] of existing benchmarks [48, 51, 77, 38] through careful filtering. Nonetheless, we report results on several classical general reasoning benchmarks [29, 42, 26, 77] in Tab. 2 for experimental completeness.

## 5.2 Main Results

As shown in Tab. 1, our VisPer-LM outperforms the single encoder baseline, i.e., LLaVA-1.5 [45] across different base encoder [14, 55] and decoder LLM [64, 65] combinations. Specifically, Llama3-8b [64] based VisPer-LM outperforms LLaVA-1.5 by **5.6%** and **4.5%** on the *Distance* and *Relation* task, respectively for the CLIP-ViT-L base encoder and by **8.7%** on the *Depth* task for the

Table 7: **Ablation on the nature of special tokens during IFT.** Keeping $\langle t \rangle$ frozen during IFT aids in keeping their task-specific nature intact, resulting in better performance.

| $\langle t \rangle$ **during IFT** | **CV-Bench$^{2D}$** | **CV-Bench$^{3D}$** | **MMStar** | **RWQA** | **Avg** |
|---|---|---|---|---|---|
| frozen | **58.6** | **64.2** | **39.5** | **57.9** | **55.1** |
| learnable | 56.9 | 56.1 | 39.0 | 57.3 | 52.3 |

CLIP-ConvNext-XXL base encoder. Furthermore, we compare our approach to the corresponding two variants of multi-encoder baselines: (i) *feat concat.*: features from all encoders are concatenated along the feature dimension and passed into a single MLP; and (ii) *token concat.*: features from all encoders are first passed through separate MLPs and then concatenated along with token dimension after average pooling the sequence outputs from $\mathbf{E}^{\text{depth}}$ and $\mathbf{E}^{\text{seg}}$ into eight tokens each. As shown in Tab. 1, our VisPer-LM outperforms the multi-encoder baselines on average, showing the effectiveness of our approach while using a single encoder during inference. Note that the multi-encoder baselines are better than VisPer-LM on the Depth task in CV-Bench owing to their superior depth representation quality as observed in Sec. 3.

We also compare our VisPer-LM with a RADIO-ViT-L [58] and Llama3-8b-based LLaVA-1.5. Unlike our approach, which distills information directly into the LLM representations, RADIO distills knowledge from multiple expert encoders into a single student encoder. As shown in Tab. 1, our VisPer-LM outperforms the RADIO-ViT-L-based LLaVA-1.5, underscoring the effectiveness of our approach over encoder distillation for developing MLLMs. Note that we train the RADIO-ViT-L-based model using the same setting as the single encoder baseline.

Additionally, we train a Llama3-8b based Cambrian-1 [67] model (with a batch size of 512, following Tong *et al.* [67]) using the LLaVA-558K (PT) plus LLaVA-665K (IFT) dataset mix. As shown in Tab. 1, the Cambrian-1 model not only underperforms as compared to our VisPer-LM but also LLaVA-1.5, underscoring the effectiveness of our approach with limited data, as the publicly released Cambrian-1 models were trained on millions of samples, requiring immense resources.

**Additional Benchmarks.** Although not the target tasks for our method, we report results on classical visual reasoning benchmarks like POPE [42], GQA [29], MMMU [77], and VizWiz [26] in Tab. 2. VisPer-LM outperforms LLaVA-1.5, demonstrating its superior visual perception ability without any loss in general reasoning abilities.

**Scalability with VPT.** We report results using the 2.5-stage training setup (PT+VPT+IFT) for LLaVA-1.5 [45] and VisPer-LM in Tab. 3. Our VisPer-LM outperforms the baselines on average across all CV-Bench tasks, highlighting the scalability of our approach with more training data.

**Results on BLINK [22] benchmark.** We report results on visual perception tasks under the BLINK benchmark (val set) in Tab. 4. We also report results using the data-matched Cambrian-1 and RADIO-ViT-L-based LLaVA-1.5 models. We find that our VisPer-LM significantly outperforms all other baselines on average, especially on the Spatial Relation task, demonstrating the effectiveness and generalization of our approach.

## 5.3 Ablations

**Layer Sets for Embedding Losses.** The choice of layers in $\mathbb{D}$, $\mathbb{S}$, and $\mathbb{G}$ for the corresponding embedding losses is a crucial design choice with a significant effect on the performance. Based on the findings about middle layers in Sec. 3, we ablate on embedding loss positions only in the middle layers of the LLM. As shown in Tab. 5, setting $\mathbb{D}$, $\mathbb{S}$, and $\mathbb{G}$ as $\{8, 20\}$, $\{10, 18\}$, and $\{12, 20\}$, respectively, performs the best overall for the 32-layer Llama3-8b.

**Training Stage for Embedding Optimization.** In Table 6, we observe that adding $\mathcal{L}_{\text{emb}}$ during IFT results in worse performance than doing so only during the PT stage. We attribute this to the interference of vision-centric supervision with task-aligned natural language supervision during IFT. Additionally, the second row of Table 6 shows that using learnable tokens in the sequence [24] without $\mathcal{L}_{\text{emb}}$ slightly improves performance over the baseline but remains inferior to VisPer-LM.

**Nature of Special tokens during IFT.** As shown in Tab. 7, keeping the special tokens ($\langle t \rangle$) frozen during IFT performs better as compared to making them learnable. We attribute frozen tokens

Table 8: **Throughput Analysis.** Our VisPer-LM has superior inference throughout compared to that of the multi-encoder baseline and similar throughput to that of the single-encoder baseline with much better performance.

| inference | single-encoder | multi-encoder | **VisPer-LM** |
|---|---|---|---|
| throughput (samples/sec) | $9.92 \pm 0.03$ | $5.32 \pm 0.02$ | $9.86 \pm 0.01$ |

Table 9: **Experiments with SigLIP-VIT-SO400M [79] as the base vision encoder.** Our VisPer-LM improves significantly over LLaVA-1.5 [45] on depth and relation reasoning, our target perception tasks.

| Method | CV-Bench | Count | Depth | Relation |
|---|---|---|---|---|
| LLaVA-1.5 | 56.3 | 47.8 | 60.3 | 57.4 |
| **VisPer-LM** (ours) | 57.3 (+1.0) | 48.7 (+0.9) | 63.7 (+3.4) | 66.3 (+8.9) |

performing better to the gradients from natural language supervision interfering with the vision-centric information stored in the special tokens during IFT if those are set as learnable parameters.

**Throughput Analysis.** We demonstrate the superior inference throughput of VisPer-LM over the multi-encoder baseline in Tab. 8. We record the throughput on a single NVIDIA 80G A100 GPU for a single forward pass on the CV-Bench evaluation set with a batch size of 1. We report the mean and standard deviation across 10 runs. We use the CLIP-ConvNeXT-XXL [14] and Llama3-8b [64] based models for the throughput analysis.

**Experiments with SigLIP [79].** In Tab. 1, we achieved performance gains with CLIP-ConvNeXT-XXL, one of the strongest visual encoders, as also found in Cambrian-1 [67] (ranked **second** overall in their Tab. 2). Here, we experiment with SigLIP-ViT-SO400M (ranked **first** in Cambrian-1 [67]) as our base visual encoder. As expected, we notice performance boosts on the perception tasks in CV-Bench in Tab. 9, which have a high positive correlation to the depth and seg representation quality (Appendix D), underscoring the effectiveness of our approach with various vision backbones.

# 6 Conclusion

In this work, we probed MLLMs and established a positive correlation between visual representation quality within the LLM and downstream performance. Building on these insights, we introduced VisPer-LM, the first approach to distill knowledge from target encoders into the LLM via predictive embedding optimization during the pre-training stage, complementing the next token prediction objective. We validated that our embedding optimization results in better vision-language alignment before the IFT stage. Through extensive experiments, we demonstrate VisPer-LM's superiority to the corresponding baselines in terms of both representation quality and VQA performance.

**Limitations and Future Work.** With VisPer-LM, we enhanced the quality of representations inside the LLM with guidance from expert visual perception encoders. That said, our work mainly focuses on improving perceptual reasoning without any loss of general reasoning abilities. Incorporating more general-purpose teacher encoders, such as InternViT [11, 12], offers a promising pathway to improve general reasoning abilities. Secondly, in this work, we mainly work with the image modality. Applying predictive embedding optimization for low-level information like motion control [4, 5] while training on videos could improve MLLMs' spatial and temporal reasoning in the future. Lastly, due to resource constraints, we could not scale VisPer-LM to larger LLMs like Llama3-70b [64] or Qwen2-72b [66], and it remains an intriguing experiment.

**Acknowledgements.** We extend our heartfelt gratitude to Fangrui Zhu, Reuben Tan, Min Shi, Bhavika Devnani, Fiona Ryan, and Chieh-Yun Chen for their valuable feedback and insightful discussions. We also sincerely thank the GCR team at Microsoft for their support in resolving frequent infrastructure challenges, which enabled our experimentation. This work was in part supported by NSF CAREER Award #2239840, and the National AI Institute for Exceptional Education (Award #2229873) by National Science Foundation and the Institute of Education Sciences, U.S. Department of Education. Lastly, we thank the ML Center @Georgia Tech and Microsoft Research for supporting this work.

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

| Method | PT | IFT | CV-Bench$^{2D}$ | CV-Bench$^{3D}$ | MMStar | OK-VQA | Avg |
|---|---|---|---|---|---|---|---|
| LLaVA-1.5 | LLaVA-558K | LLaVA-665k | 60.0 | 56.3 | 37.4 | 56.0 | 52.4 |
| LLaVA-1.5 | LLaVA-558K + ALLaVA-Caption-663K | LLaVA-665k | 56.8 | 60.8 | 37.1 | 57.5 | 53.1 |
| **VisPer-LM** | LLaVA-558K | LLaVA-665k | **60.8** | **62.2** | **38.5** | **59.0** | **55.1** |

Table II: Setting $\mathbf{N} = 8$ is optimal with $\mathbf{N} = 0$ setting also giving boosts on CV-Bench and MMStar.

| N | CV-Bench$^{2D}$ | CV-Bench$^{3D}$ | MMStar | RWQA | Avg |
|---|---|---|---|---|---|
| single-encoder | 56.0 | 61.0 | 38.8 | 57.8 | 53.4 |
| 0 | 56.1 | 62.0 | **40.1** | 56.3 | 53.6 |
| 8 | **58.6** | **64.2** | 39.5 | **57.9** | **55.1** |
| 16 | 56.6 | 63.6 | 37.1 | 54.5 | 52.9 |
| 24 | 55.7 | 60.0 | 39.3 | 57.4 | 53.1 |

Table III: **Key input to the Embedding Predictor.**. Feeding the tokens corresponding to the system prompt, image, corresponding special tokens, and the text query is optimal.

| key input to emb. predictor | CV-Bench$^{2D}$ | CV-Bench$^{3D}$ | MMStar | RWQA | Avg |
|---|---|---|---|---|---|
| $\langle img \rangle \mid \langle t \rangle$ | 53.0 | 54.6 | 38.4 | 56.7 | 50.7 |
| $\langle sys \rangle \mid \langle img \rangle \mid \langle t \rangle$ | **58.7** | 63.0 | 38.8 | 57.4 | 54.5 |
| $\langle sys \rangle \mid \langle img \rangle \mid \langle t \rangle \mid \langle txt \rangle$ | 58.6 | **64.2** | **39.5** | **57.9** | **55.1** |

# Appendix

In this appendix, we first share additional ablations, including the effect of the order of different special tokens ($\langle g \rangle, \langle d \rangle, \langle s \rangle$) in the input sequence to the LLM and the different key input possibilities to the embedding predictor in Appendix A. We use CLIP-ViT-L [55] and Llama3-8b [64] as the base vision encoder and decoder LLM, respectively, for the ablations, unless mentioned otherwise. Next, we demonstrate the effectiveness of our probing setup for encoder-free MLLMs in Appendix C and establish a clear correlation between probing performance and CV-Bench accuracy in Appendix D. Lastly, we provide qualitative and quantitative analysis on the downstream probing tasks in Appendix E.

## A    Additional Ablations

**Number of Special Tokens.** We analyze the effect of the number of special tokens per target feature ($\langle t \rangle$) on the performance in Tab. II. Setting $\mathbf{N}$ as eight results in the best average performance across benchmarks. Furthermore, VisPer-LM without $\langle t \rangle$ also outperforms the baseline on CV-Bench$^{3D}$ and MMStar, respectively, demonstrating the effectiveness of our embedding optimization.

**Input tokens to Embedding Predictor.** As shown in Tab. III, we find that including the tokens corresponding to the system prompt in the key input to the embedding predictor is critical for performance. We attribute it to system tokens having high attention scores and effect on the generation [73]. Therefore, distilling target information into the system tokens is crucial for performance. Moreover, including the text query tokens in the key input to the embedding predictors also results in a slight performance boost as the text tokens hold global image information [33].

**Embedding Optimization Mode.** In this ablation study, we evaluate various combinations of embedding losses applied during pretraining (PT). Our results, summarized in Tab. IV, reveal that the optimal performance is achieved when all three embedding losses—depth, seg, and gen—are used together. Interestingly, we observe that utilizing only depth or gen embedding losses still leads to notable performance improvements, whereas relying solely on seg embedding loss does not yield significant gains. This suggests that different types of target information contribute uniquely to the distillation process. Investigating how the distillation of one type of target information influences the effectiveness of others presents an intriguing direction for future research.

Table IV: **Embedding Optimization Modes.** Using the depth, seg, and gen embedding losses simultaneously is optimal.

| mode | CV-Bench$^{2D}$ | CV-Bench$^{3D}$ | MMStar | Avg |
|---|---|---|---|---|
| LLaVA-1.5 | 56.0 | 61.0 | 38.8 | 51.9 |
| depth | **58.6** | 63.5 | 38.8 | 53.6 |
| seg | 56.2 | 57.6 | 38.2 | 50.7 |
| gen | 56.2 | **65.8** | 39.3 | 53.8 |
| depth + seg | 58.6 | 61.8 | 38.6 | 53.0 |
| depth + gen | 53.6 | 61.8 | 38.8 | 51.4 |
| seg + gen | 54.2 | 60.2 | 39.3 | 51.2 |
| depth + seg + gen | **58.6** | 64.2 | **39.5** | **54.1** |

Table V: **Order of different special tokens in the input sequence to the LLM.** Appending the gen, depth, and seg tokens (in that order) in the LLM's input sequence after the image tokens is the optimal setup.

| $\langle t \rangle$ order | Count$^{2D}$ | Depth$^{3D}$ | Relation$^{2D}$ | Distance$^{3D}$ | Overall |
|---|---|---|---|---|---|
| LLaVA-1.5 | 50.4 | 73.3 | 64.9 | 48.7 | 58.5 |
| $\langle d \rangle \mid \langle s \rangle \mid \langle g \rangle$ | 49.4 | 68.7 | 69.2 | **56.2** | 59.9 |
| $\langle d \rangle \mid \langle g \rangle \mid \langle s \rangle$ | **51.6** | 72.8 | 70.3 | 54.5 | **61.4** |
| $\langle s \rangle \mid \langle d \rangle \mid \langle g \rangle$ | 48.7 | 71.3 | 65.2 | 52.5 | 58.5 |
| $\langle s \rangle \mid \langle g \rangle \mid \langle d \rangle$ | 46.7 | 71.3 | **71.2** | 50.8 | 58.9 |
| $\langle g \rangle \mid \langle d \rangle \mid \langle s \rangle$ | 51.3 | **74.2** | 69.4 | 54.3 | **61.4** |
| $\langle g \rangle \mid \langle s \rangle \mid \langle d \rangle$ | 50.9 | 68.8 | 70.0 | 50.5 | 59.2 |

Table VI: **Embedding Loss weights during PT.** Setting each embedding loss' weight to 0.5 is optimal.

| $\lambda_{\text{depth}}$ | $\lambda_{\text{seg}}$ | $\lambda_{\text{gen}}$ | CV-Bench$^{2D}$ | CV-Bench$^{3D}$ | MMStar | Avg |
|---|---|---|---|---|---|---|
| LLaVA-1.5 | | | 56.0 | 61.0 | 38.8 | 51.9 |
| 0.10 | 0.10 | 0.10 | **60.5** | 61.3 | 38.3 | 53.4 |
| 0.25 | 0.25 | 0.25 | 56.3 | 59.4 | 37.1 | 50.9 |
| 0.50 | 0.50 | 0.50 | 58.6 | **64.2** | **39.5** | **54.1** |
| 0.75 | 0.75 | 0.75 | 57.9 | 59.4 | 37.6 | 51.6 |
| 1.00 | 1.00 | 1.00 | 55.8 | 61.7 | 38.1 | 51.9 |

Table VII: **Ablations on components of embedding losses.** Using both smooth-L1-loss and contrastive loss to compute the final embedding loss is optimal.

| $\mathcal{L}_{\text{sL1}}$ | $\mathcal{L}_{\text{contrastive}}$ | CV-Bench$^{2D}$ | CV-Bench$^{3D}$ | MMStar | Avg |
|---|---|---|---|---|---|
| ✓ | | 56.8 | 62.3 | 38.3 | 52.5 |
| ✓ | ✓ | **58.6** | **64.2** | **39.5** | **54.1** |

**Order of Special Tokens.** In Tab. V, we ablate the order of different special tokens in the LLM's input sequence. We find that $\{\langle g \rangle \mid \langle d \rangle \mid \langle s \rangle\}$ and $\{\langle d \rangle \mid \langle g \rangle \mid \langle s \rangle\}$ show the best performance on CV-Bench [67]. We choose $\{\langle g \rangle \mid \langle d \rangle \mid \langle s \rangle\}$ as our default order due to its performance being better than the baseline on all sub-tasks in CV-Bench.

**Embedding Loss weights.** In Tab. VI, we ablate on different values of $\lambda_{\text{depth}}$, $\lambda_{\text{seg}}$, and $\lambda_{\text{gen}}$ during for the corresponding embedding losses during the pre-training stage. We find that setting each loss weight to 0.5 is optimal.

**Effect of Contrastive Embedding Loss.** In Tab. VII, analyze the impact of the contrastive loss component within the embedding loss. Our findings show that incorporating the contrastive loss significantly enhances performance, highlighting its positive influence on the model's effectiveness.

Table VIII: **Enhanced Projector post Predictive Embedding Optimization.** We observe that the projector features in VisPer-LM are better than those in LLaVA-1.5 after the PT stage (`proj_PT`), indicating a positive effect of the embedding losses during the PT stage on the learned projector. We use CLIP-ConvNeXT-XXL and Llama3-8B as the base visual encoder and decoder LLM, respectively.

| Category | Model | Module Output Probed | Stage | depth | seg | gen | avg |
|---|---|---|---|---|---|---|---|
| `vision_encoder` | CLIP-ConvNeXT-XXL | | — | 0.545 | 0.512 | 0.719 | 0.592 |
| `proj_PT` | LLaVA-1.5 | Projector | PT | 0.531 | 0.495 | **0.781** | 0.602 |
| | **VisPer-LM** (ours) | Projector | PT | **0.554** | **0.503** | 0.776 | **0.611** |

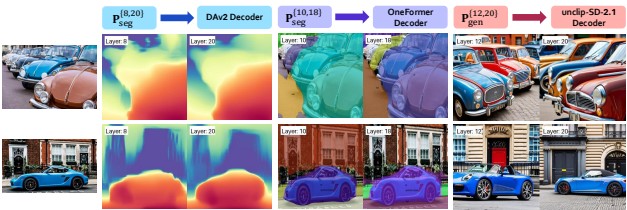

Figure I: **Visualizing Embedding Predictor Outputs after the PT stage.** The quality of the decoded representations indicates the effectiveness of our embedding optimization.

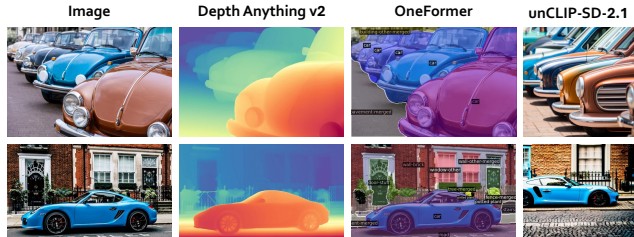

Figure II: Ground-truth outputs from the target models used for Probing MLLMs.

Table IX: **Evaluating Encoder-Free MLLMs on CV-Bench.** Our VisPer-LM significantly outperforms native encoder free VLMs like EVE-7B [19] and Chameleon-7B [63].

| Method | Visual Encoder | LLM | Count[2D] | Depth[3D] | Relation[2D] | Distance[3D] | Overall |
|---|---|---|---|---|---|---|---|
| Chameleon [63] | — | Llama2-7b | 13.6 | 51.7 | 49.5 | 50.8 | 39.6 |
| EVE [19] | — | Vicuna-7b | 52.2 | 61.3 | 69.2 | 53.8 | 58.4 |
| LLaVA-1.5 | CLIP-ConvNeXT-XXL | Llama3-8b | 54.1 | 62.8 | **69.5** | 49.8 | 58.2 |
| **VisPer-LM** (ours) | CLIP-ConvNeXT-XXL | Llama3-8b | **57.4** | **71.5** | 66.8 | **52.8** | **61.5** |

We keep the smooth L1 loss as a default component to ensure the embedding predictions maintain the same magnitude as the target features, which is crucial for meaningful visualization.

**Qualitative Comparisons.** We provide qualitative comparisons demonstrating the difference between LLaVA-1.5 [45] and **VisPer-LM** for the different tasks in CV-Bench in Fig. VI, Fig. VII, Fig. VIII, and Fig. IX.

**Visualizing Embedding Predictions for Target Tasks.** We visualize the visual quality of LLM representations after the PT stage for our VisPer-LM using the decoders from the corresponding target models, as shown in Fig. I. We observe that the decoder outputs have good object shape and boundary quality, demonstrating the successful representation optimization with our embedding losses.

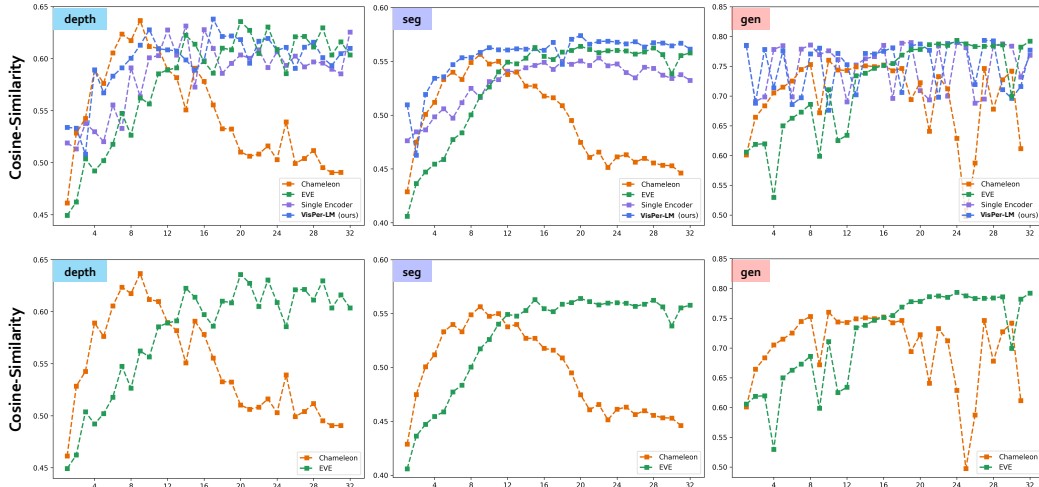

Figure III: **Probing Encoder-Free MLLMs.** We probe Chameleon-7B [63] and EVE-7B [19] models to show the effectiveness of our probing setup for encoder-free MLLMs. In the first row, we compare the mentioned models to LLaVA-1.5 and our VisPer-LM from Sec. 3 of the main text. On the one hand, we notice that EVE-7B shows worse probing performance in the initial layers than LLaVA-1.5, and the representation quality follows an upward trend even in deeper layers, resulting in features of similar quality to LLaVA-1.5, which is also reflected in the CV-Bench accuracy. On the other hand, Chameleon-7B has the best representation quality in the early middle layers, with the quality dropping significantly in the deeper layers. We include only the EVE-7B and Chameleon in the second row for better readability.

## B    Enhanced Projector post Pre-Training

In this section, we probe the features from the visual encoder and the projector for the LLaVA-1.5 and VisPer-LM models from Sec. 3 of the main text, i.e., using CLIP-ConvNeXT-XXL and the Llama3-8B as the base visual encoder and decoder LLM, respectively. Note that only the projector is trained during the PT stage, while the projector and the LLM are trained during the IFT stage. The visual encoder is kept frozen during both stages.

We compare the probing scores for the feature outputs from the projector module of the LLaVA-1.5 and VisPer-LM models between checkpoints obtained after the PT stage (`proj_PT`) of training. As shown in Tab. VIII, the representations in models under `proj_PT` are better for VisPer-LM than LLaVA-1.5, indicating the predictive embedding optimization assisting in learning an improved projector before the IFT stage.

We observe that the quality of depth and seg representations in LLaVA-1.5 under `proj_PT` are worse than those of the base visual encoder (`vision_encoder`), indicating a loss of depth/spatial information after the features are fed into the projector in LLaVA-1.5. However, for VisPer-LM, the depth representations have better quality than those from the vision encoder. Note that OLA=VLM's seg representations are slightly worse than those from CLIP-ConvNeXT-XXL but still better than those from LLaVA-1.5.

## C    Probing Encoder-Free MLLMs

Following the settings described in Sec. 3 of the main text, we probe all layers from the publicly available encoder-free Chameleon-7B [63] and EVE-7B [19] model checkpoints against the depth, seg, and gen target features in Fig. III.

We notice that the representation quality for features from EVE-7B is relatively worse in the initial layers, owing to the patch embedding operation that outputs coarse features. However, the probing performance improves in the deeper layers, which could be attributed to the patch-alignment operation [19]. Notably, EVE-7B uses about 35M [19] training samples but still lags slightly behind

Table X: **Correlation of CV-Bench performance to probing performance.** We notice the high positive correlation of 0.98 between depth probing performance and CV-Bench accuracy. Moreover, we find that performance on the Count and Depth subtasks also highly correlates with the seg and depth probe cosine-similarity scores, respectively. Additionally, we find a low correlation between the gen probing performance and CV-Bench performance.

| Probe Mode | Count$^{2D}$ | Depth$^{3D}$ | Relation$^{2D}$ | Distance$^{3D}$ | CV-Bench$^{2D}$ | CV-Bench$^{3D}$ | CV-Bench |
|---|---|---|---|---|---|---|---|
| depth | 0.90 | **0.98** | **0.70** | -0.31 | **0.96** | 0.94 | **0.98** |
| seg | **0.98** | 0.93 | 0.50 | **-0.15** | 0.91 | **0.96** | **0.98** |
| gen | 0.34 | 0.46 | 0.27 | -0.31 | 0.37 | 0.36 | 0.37 |

LLaVA-1.5 in performance (Tab. IX), with our VisPer-LM showing better representation quality and performance, underscoring the effectiveness of our approach.

Unlike EVE-7B, features from Chameleon-7B [63] have surprisingly low depth and seg representation quality in the deeper layers. Unlike other models, Chameleon-7B follows a steep downward trend in the deeper layers, which we attribute to its relatively low training on VQA data with a heavy focus on text-to-text and text-to-image tasks in the SFT data mixture, which is also evident from its low performance on CV-Bench. Note that during evaluation, we noticed that Chameleon cannot output *option letters* required to evaluate on MCQ-type CV-Bench. Therefore, we leverage Qwen-2.5-72B-Instruct [66] to extract the option letters corresponding to the sentence output from Chameleon to obtain the numbers for Tab. IX.

For both the models, gen probes perform relatively better due to the text-aligned nature of the gen target features.

These findings provide critical insights into developing future encoder-free MLLMs, re-establishing the importance of VQA data in the SFT mixture, and supervising the vision tokens inside the LLM from an expert encoder during training. Our proposed embedding optimization opens up a promising avenue to this end.

## D   Correlation between VQA Performance and Probing Performance

> **Takeaways**
>
> **1.** As shown in Tab. X, we observe the highest positive correlation between CV-Bench accuracy and seg/depth probing performance.
> **2.** Moreover, we find a high correlation between the performance on the *Depth/Relation* task and *Count* tasks with the depth and seg probing performance, respectively.
> **3.** As expected, we find low performance correlation with gen probe scores, indicating that gen features act as a good proxy for general visual representation targets.
> **4.** Surprisingly, we observe a negative correlation between the Distance task and all probe scores, which could be attributed to the lack of "real-world distance" estimation queries in the training dataset.

In this section, we analyze the correlation between the performance on CV-Bench (evaluates models' visual perception ability) and probing performance (measure of model's representation quality), basing our conclusion on trends for six models and their corresponding depth/seg/gen probes:

- **LLaVA-1.5-8B-50%-IFT**: A LLaVA-1.5 model trained with complete pre-training (PT) but only 50% completion of the instruction fine-tuning (IFT) stage. It employs CLIP-ConvNeXT-XXL [14, 50] as the visual encoder and Llama3-8B [64] as the decoder LLM.
- **LLaVA-1.5-8B**: A LLaVA-1.5 model trained with complete PT+IFT stages. It employs CLIP-ConvNeXT-XXL [14, 50] as the visual encoder and Llama3-8B [64] as the decoder LLM.
- **VisPer-LM-8B**: Our VisPer-LM model trained with complete PT+IFT stages while using predictive embedding optimization during the PT stage. It employs CLIP-ConvNeXT-XXL [14, 50] as the visual encoder and Llama3-8B [64] as the decoder LLM.

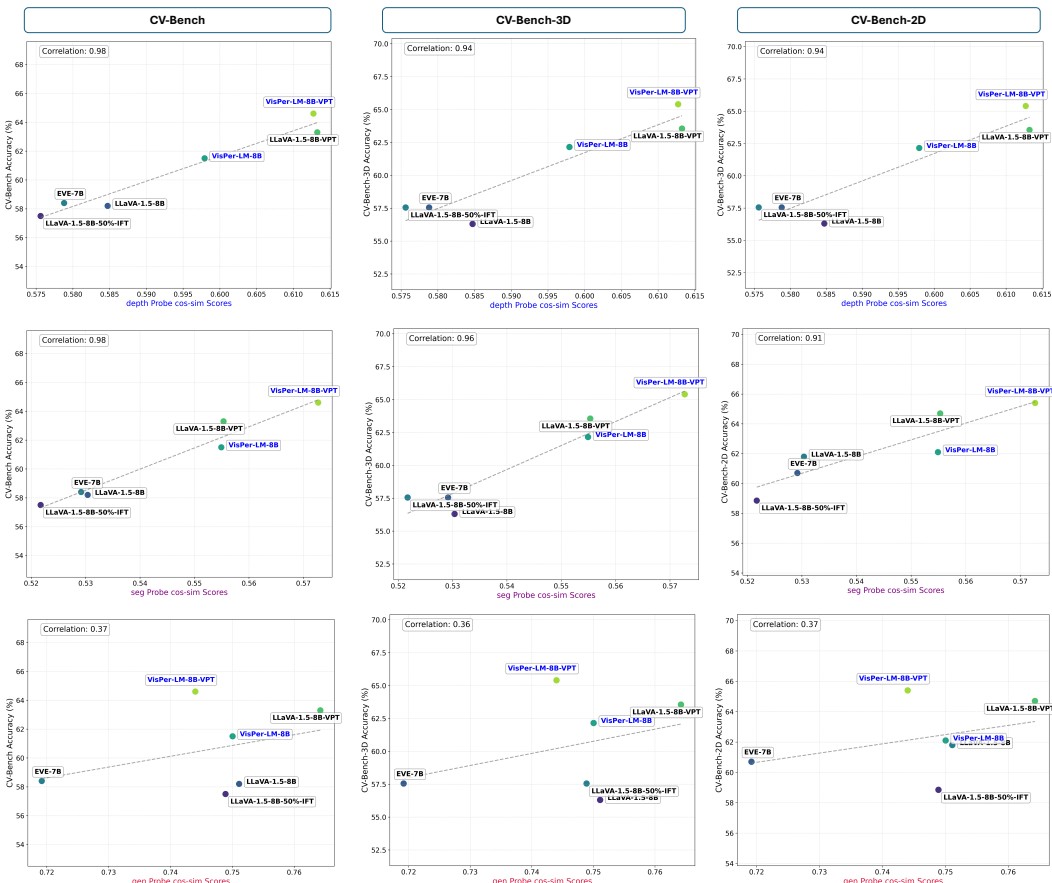

Figure IV: **Plotting Correlation between Probing performance and CV-Bench accuracy.** We find a high correlation between the eval cosine-sim scores and the accuracies on the CV-Bench as well as the 2D and 3D categories under CV-Bench for the depth and seg probes, validating their choice as the target task to improve the MLLM's perception ability.

- **LLaVA-1.5-8B-VPT**: A LLaVA-1.5 model trained with complete PT+VPT+IFT stages. It employs CLIP-ConvNeXT-XXL [14, 50] as the visual encoder and Llama3-8B [64] as the decoder LLM.
- **VisPer-LM-8B-VPT**: Our VisPer-LM model trained with complete PT+VPT+IFT stages using predictive embedding optimization during the PT stage. It employs CLIP-ConvNeXT-XXL [14, 50] as the visual encoder and Llama3-8B [64] as the decoder LLM.
- **EVE-7B**: The publicly available EVE-7B model checkpoint from [19]. It does not use a visual encoder and is built on Vicuna-7B [15]

We follow the same settings as mentioned in Sec. 3 and Sec. 5 of the main text to train our probes and models, respectively.

**Probing Mode: depth.** We observe a high correlation of 0.98 between the CV-Bench accuracy and depth probe eval cosine-sim scores, as shown in Fig. IV. It is critical proof of our claim that improved visual representations and the corresponding VQA performance go hand-in-hand. We also find a correlation with a 0.98 and 0.80 factor between the accuracy on the Depth and Relation tasks (in CV-bench), respectively, with the depth probe scores in Fig. V, explaining the gains achieved by VisPer-LM.

**Probing Mode: seg.** We also observe a high correlation of 0.98 between the CV-Bench accuracy and seg probe eval cosine-sim scores, as shown in Fig. IV. The high correlation indicates that CV-Bench requires strong perception abilities, validating our decision to adopt it as the primary benchmark for our work. We also find a correlation of 0.98 between the accuracy on the Count task (in CV-bench) with the seg probe scores in Fig. V.

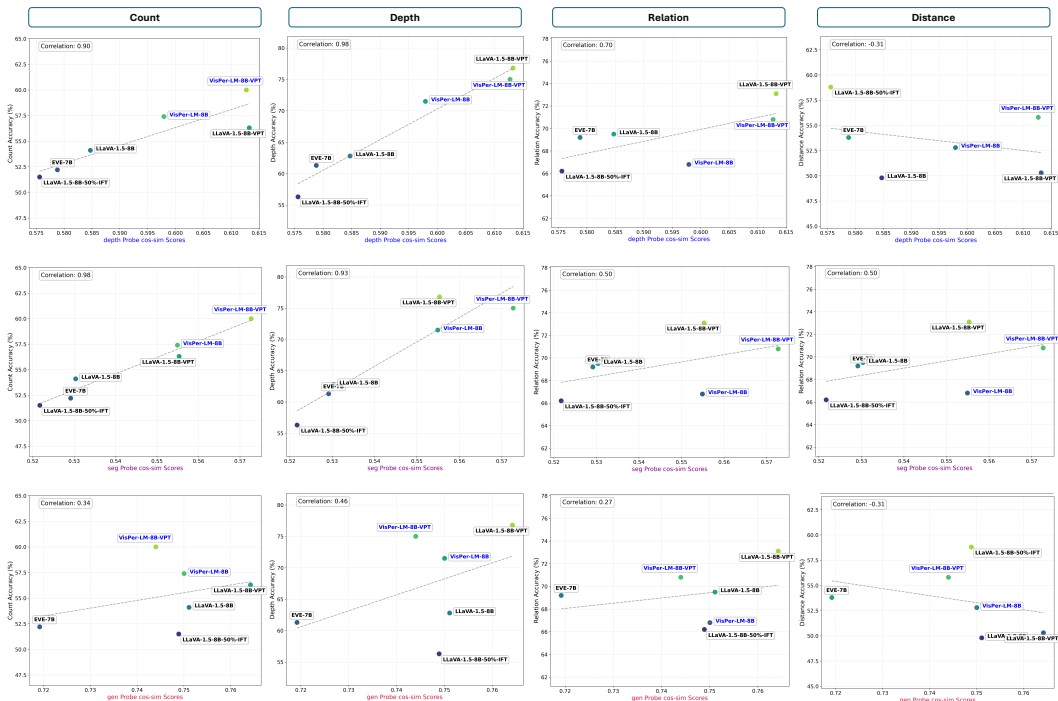

Figure V: **Plotting Correlation between Probing performance and CV-Bench sub-tasks accuracy.** We find a high correlation between the depth eval cosine-sim scores and the Depth/Relation/Count task performances. We make the same observation about seg probe scores and the Count/Depth task. Interestingly, we find a low negative correlation between the probe scores and Distance task performance.

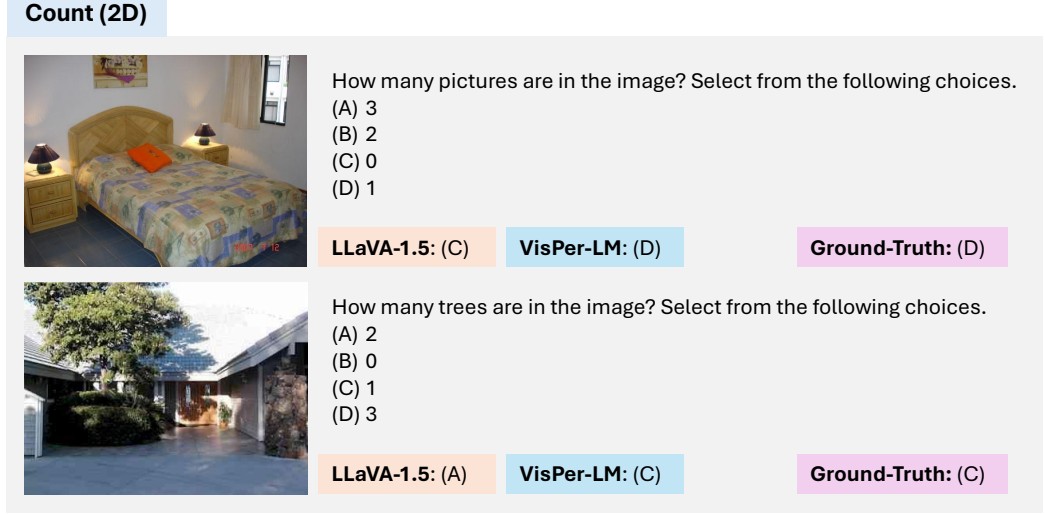

Figure VI: **Qualitative Examples for the Count task in CV-Bench.** Our VisPer-LM can accurately predict the presence of one picture and one tree, unlike LLaVA-1.5 [45].

**Probing Mode: gen.** We observe that the gen probe scores do not correlate highly with CV-Bench or any of its tasks, as shown in Fig. IV and Fig. V. We attribute this finding to the generative features' text-aligned nature, which is not optimal for developing the model's perception abilities.

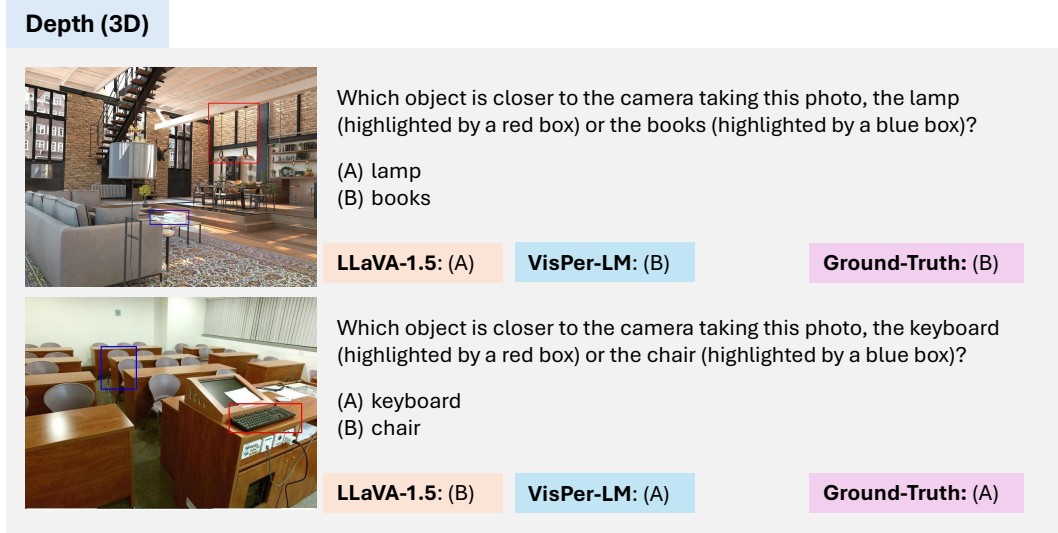

Figure VII: **Qualitative Examples for the Depth task in CV-Bench.** Our VisPer-LM can accurately predict that the lamp and keyboard ar closer to the camera in the respective samples.

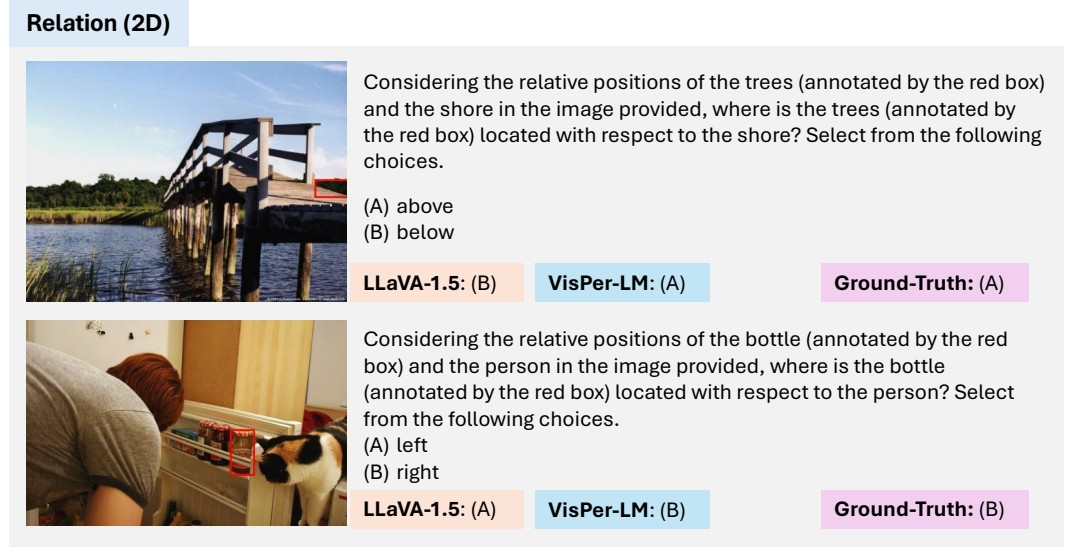

Figure VIII: **Qualitative Examples for the Relation task in CV-Bench.** Our VisPer-LM can accurately predict that the positions of the trees and the bottle in the respective samples.

## E    Evaluating Probes on Downstream Tasks

We evaluate the probes trained against the target features on the corresponding downstream target tasks, i.e., image generation, depth estimation and image segmentation. To obtain the predictions, we feed the outputs from the probes into the decoder from the corresponding target model. Specifically, we report the FID [60] scores on 5k images from COCO-val2017 [43] for gen probes, accuracy on the DA-2K [75] benchmark for depth probes, and mIoU [31] on the COCO-val2017 [43] set for seg probes. We average the scores over all layers for easier comparison. As shown in Tab. XI, probes trained for our VisPer-LM outperform those for the baseline LLaVA-1.5 [45] model across all probing tasks, proving the improved visual representation quality owing to our embedding optimization approach.

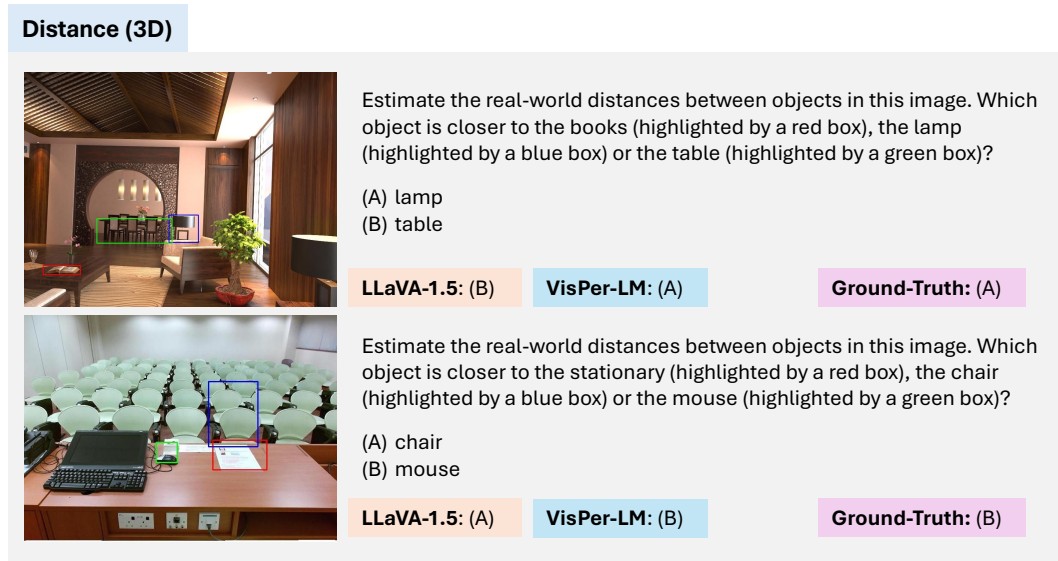

Figure IX: **Qualitative Examples for the Distance task in CV-Bench.** Our VisPer-LM can accurately predict that the distances between the respective pair of objects.

Table XI: **Quantitative Evaluation on target probing task.** Probes trained for our VisPer-LM perform significantly better as compared to the probes trained on baseline LLaVA-1.5 [45].

| Probed Model | FID [60] (↓) | DA-2K % Acc. [75] (↑) | % mIoU [31] (↑) |
|---|---|---|---|
| LLaVA-1.5 | 23.1 | 66.4 | 39.3 |
| **VisPer-LM** (ours) | **22.4** | **77.8** | **45.4** |
| Target Encoder | 18.1 | 97.3 | 64.5 |

|  LLaVA-1.5  |  **VisPer-LM** (ours) |
|---|---|

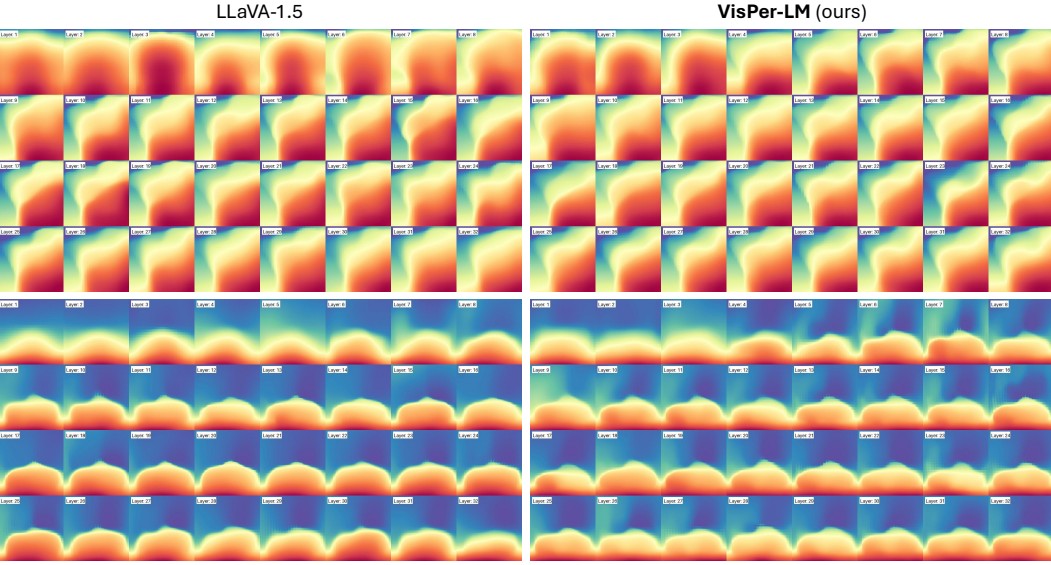

Figure X: **Layerwise visualizations for the depth probes.** For LLaVA-1.5 [45], the probes generate blob-like outputs up to the eighth layer, with visualizations progressively improving in the middle layers, aligning with the findings presented in Sec. 3 of the main text. Notably, probes trained on VisPer-LM begin producing distinguishable object shapes and boundaries as early as the third layer, attributed to the incorporation of embedding losses leading to improved representations in the initial layers.

LLaVA-1.5                                    **VisPer-LM** (ours)

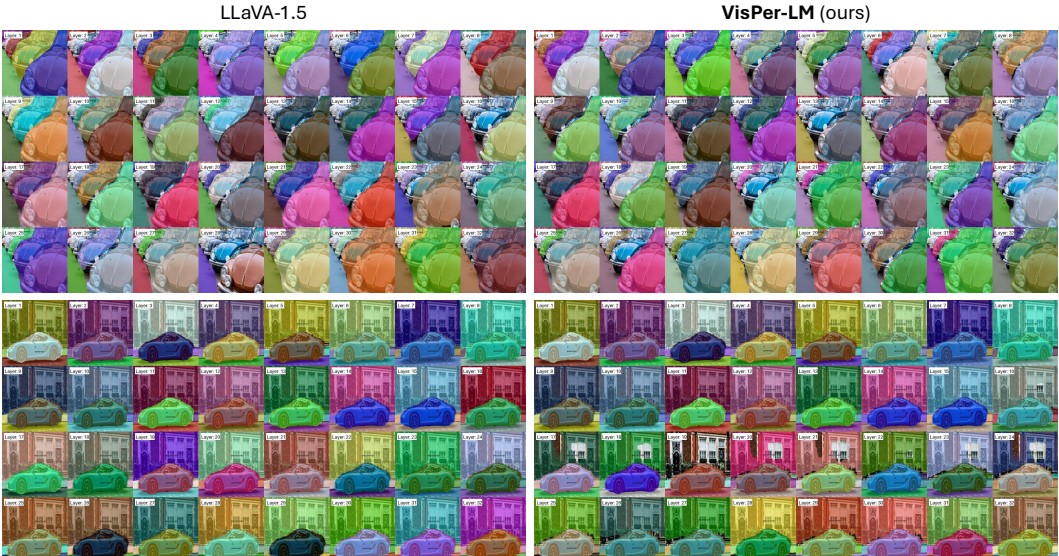

Figure XI: **Layerwise visualizations for the seg probes.** The LLaVA-1.5 probes often fail to segment the third car in the background for the first sample during the initial layers (layers two to eight), whereas the VisPer-LM probes demonstrate relatively better performance in this scenario.

LLaVA-1.5                                    **VisPer-LM** (ours)

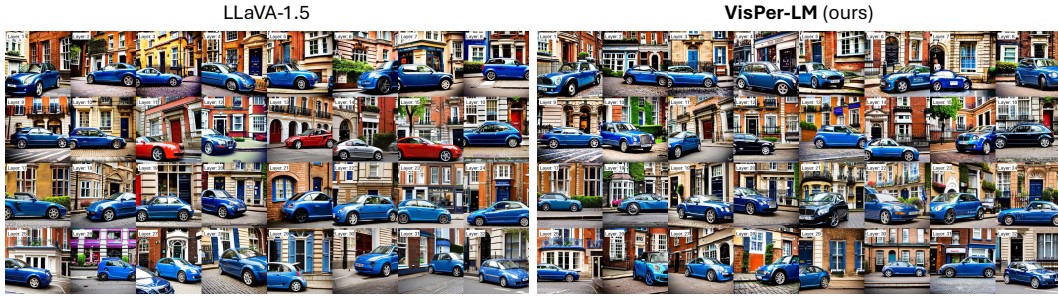

Figure XII: **Layerwise visualizations for the gen probes.** The probe outputs for both the models are of fairly good quality.

**Visualizing Probe Outputs.** We present visualizations for the outputs for the probes trained on the single-encoder LLaVA-1.5 model in Sec. 3 in the main text. We provide the ground-truth visualizations from the teacher models in Fig. II.

As shown in Fig. X, the probe visualizations for the first eight layers of LLaVA-1.5 exhibit blob-like patterns, while the later layers progressively enhance the shape and boundary details of the foreground objects. In contrast, the probe visualizations for our VisPer-LM demonstrate improved object shapes and boundaries starting as early as layer-4, consistent with the findings on representation quality and layer-wise trends discussed in Sec. 3 of the main text. Additionally, we present probe visualizations for the seg and gen representations in Fig. XI and Fig. XII, respectively.

