# OpenReview forum: "Elevating Visual Perception in Multimodal LLMs with Visual Embedding Distillation"
_NeurIPS.cc/2025/Conference — NeurIPS 2025 poster_

### Official Review · Reviewer_JZQJ · 2025-06-03

**Clarity:** 4
**Significance:** 4
**Originality:** 4
**Rating:** 4
**Confidence:** 4

**Summary:**

The paper introduces VisPer-LM to enhance the visual perception of MLLMs. The model distill expert visual knowledge directly into the hidden representations of the LLM. VisPer-LM optimizes a predictive embedding loss from auxiliary vision encoders trained on segmentation, depth, and generation tasks. Extensive experiments demonstrate that VisPer-LM outperforms both single-encoder and multi-encoder baselines.

**Questions:**

I don't have further questions.

**Ethical Concerns:**

["NO or VERY MINOR ethics concerns only"]

**Final Justification:**

The rebuttal has addressed my concerns so my final rating is 4.

**Limitations:**

Yes

**Quality:**

4

**Strengths And Weaknesses:**

Strength:

1. The task is meaningful. It's important for LLMs to use visual input more precisely.

2. The proposed method uses different expert visual encoders. This new training pipeline shows good results on various benchmarks.

3. The paper is well written and easy to follow.


Weakness:

The proposed model and pipeline look promising but more experiments need to be performed, especially for spatial/depth reasoning. Please try benchmark in SpatialRGPT and SpatialBench:
https://github.com/AnjieCheng/SpatialRGPT
https://huggingface.co/datasets/RussRobin/SpatialBench

Also, comparing with LLaVA seems to be unfair, because LLaVA is based on RGB image encoder only, and is not specifically trained for spatial/depth understanding and reasoning. Please also compare with: SpatialVLM, SpatialRGPT and SpatialBot.

In all, the model and pipeline is good, but the paper lacks suuficient experiments. I'm more than happy to raise my ratings if the authors can provide more experiment results in the revised manuscript.

---

> ### Author Rebuttal · Authors · 2025-07-28
>
> We thank the reviewer for their feedback and comments.
> We are encouraged that they find our task as "*meaningful"* and *"important"*. We are glad that they acknowledge our method's *"good results on various benchmarks."* and that they find our paper easy to follow and well-writen.
>
> We respond to their questions and concerns below:
>
> ### **Pt #1: More Benchmarks**
>
> > Reviewer's Comment: The proposed model and pipeline look promising but more experiments need to be performed, especially for spatial/depth reasoning. Please try benchmark in SpatialRGPT and SpatialBench
>
> Thank you for your suggestions!
>
> We provide results on the `positional` and `counting` tasks (which are aligned with our target tasks) under SpatialBench [2]. As shown in the table below, our VisPer-LM outperforms the baseline LLaVA-1.5 model and both variants of SpatialBot (RGB and RGBD inputs). For SpatialBot, we report results as mentioned by the authors under issue #13 of the official repo due to an update of the model's checkpoint since its publication.
>
> | method                       |   positional                 |   counting            |
> |-----------------------|:----------------------------:|:----------------------------:|
> | LLaVA-1.5             |            55.9              | 86.3 |
> | SpatialBot (RGB)  |          50.0               | 87.4 |
> | SpatialBot (RGBD)  |          **61.8**              | 87.4 |
> | **VisPer-LM (ours)**  |          **61.8**               | **89.1** |
>
> We could not evaluate our model on SpatialRGPT-Bench [1] because the benchmark is designed for grounded spatial reasoning, i.e., it requires the model to be trained with region-aware referring question-answer data which is not present in our LLaVA-1.5 based training set, making it hard for the model to understand and answer the benchmark's questions in way that could help the evaluation of its abilties. We hope the reviewer understands the special nature of this benchmark and our inability to evaluate on it without a special training recipe.
>
> [1] Cheng, An-Chieh et al., "SpatialRGPT: Grounded Spatial Reasoning in Vision-Language Models." NeurIPS 2024
>
> [2] Cai, Wenxiao et al., "SpatialBot: Precise Spatial Understanding with Vision Language Models." ICRA 2025
>
>
> ### **Pt #2: Comparison to MLLMs with non-RGB encoders**
>
> > Reviewer's Comment: Also, comparing with LLaVA seems to be unfair, because LLaVA is based on RGB image encoder only, and is not specifically trained for spatial/depth understanding and reasoning. Please also compare with: SpatialVLM, SpatialRGPT and SpatialBot.
>
> We respectfully disagree that comparing with LLaVA is unfair just because it only uses an RGB image as input. Most existing MLLM works use RGB encoders including Cambrian-1 [66] which proposed CV-Bench (including the spatial and depth reasoning tasks) for evaluating MLLM's visual perception abilities. We believe it is critical to develop general-purpose MLLMs that are accurate at depth and spatial reasoning with only an RGB image input and our VisPer-LM is a significant advancement in that direction.
>
> Nonetheless, we thank the reviewer for their suggestion and provide comparisons to SpatialRGPT and SpatialBot on the Depth and Relation tasks under CV-Bench below. Interestingly, although our VisPerLM does not show the best performance on either task, it performs best on average. SpatialBot shows the best performance on the Depth task while SpatialRGPT shows the best performance on Relation task but their performance lags on the other task. We believe a plausible reason for this could be the large scale data used by these models, making them specialized for particular tasks despite using additional non-RGB inputs. We could not compare our approach to SpatialVLM [1] since the authors of that paper do not release their code, models or benchmark.
>
> | Method                | Training Data | Depth | Relation | Avg  |
> |-----------------------|:-------------:|:-----:|:--------:|:----:|
> | LLaVA-1.5             |     ~1.2M     | 70.8  |   74.0   | 72.4 |
> | SpatialBot            |     ~2.7M     | **77.2**  |   70.0   | 73.6 |
> | SpatialRGPT           |     >50M      | 59.1  |   **80.3**   | 69.7 |
> | **VisPer-LM (ours)**  |     ~1.2M     | 72.5  |   77.2   | **74.9** |
>
> [1] Chen, Boyuan et al., "SpatialVLM: Endowing Vision-Language Models with Spatial Reasoning Capabilities." CVPR 2024
>
> [66] Shengbang Tong, Ellis Brown, Penghao Wu, Sanghyun Woo, Manoj Middepogu, Sai Charitha Akula, Jihan Yang, Shusheng Yang, Adithya Iyer, Xichen Pan, Austin Wang, Rob Fergus, Yann LeCun, and Saining Xie. Cambrian-1: A fully open, vision-centric exploration of multimodal llms, NeurIPS 2024.

---

> > ### Comment · Reviewer_JZQJ · 2025-08-01
> >
> > Thank you for the reply and providing further experimental results. My concerns have been largely addressed.

---

> > > ### Author Response · Authors · 2025-08-06
> > >
> > > We thank the reviewer for their reply and time! We are glad that our rebuttal helped address their concerns. We are happy to respond to any remaining concerns, should they arise.

---

### Official Review · Reviewer_Khkk · 2025-06-30

**Clarity:** 3
**Significance:** 3
**Originality:** 3
**Rating:** 4
**Confidence:** 3

**Summary:**

This paper introduces VisPer-LM, a Multimodal LLM pre-trained with visual embedding distillation from multiple vision experts to enhance visual perception. VisPer-LM follows the established structure Vision Encoder $\rightarrow$ MLP $\rightarrow$ LLM.

The work starts with a preliminary analysis, examining whether representations within the LLM decoder can be probed to visual representations of different vision experts (depth, text-to-image, and segmentation models). To achieve this, it utilizes a Resampler block, which enables control over the output sequence length through learnable queries, and experiments with a single- and multi-encoder MLLM design. The analysis shows that different layers lead to different probing performance and, more importantly, that probing performance positively correlates with performance on CV-Bench, a vision-centric benchmark.

This motivates the rest of the work in designing a pretraining scheme to distill visual representations from vision experts to (subsets of) LLM blocks. The approach utilizes the probing setup described above and optimizes both a Smooth-L1 objective and an InfoNCE objective to align Resampler outputs with those of experts, pairing these objectives with the standard language modeling loss. Following the LLaVA recipe, only the MLP is trained during this phase (alongside the Resampler heads and some auxiliary task tokens).  Empirical results show that further using this objective during Instruction Tuning hurts performance.

The paper evaluates vision-centric abilities on CV-Bench, as well as standard multimodal abilities, showing improved visual perception with little to no degradation on common academic benchmarks.

**Questions:**

For a productive rebuttal, I would start expanding the weaknesses above into the following questions:

1. How much does the downstream performance of VisPer-LM suffer from informed decisions based on CV-Bench? Does improved visual perception persist when evaluating on other vision-centric benchmarks?
2. Did the authors experiment with other probing setups (*.e.g.* omitting textual tokens as well as / or tokens from the system prompt)?
2. About the RADIO-based LLaVA: did the authors train it themselves, or does it come from some previous work I am unaware of?
3. Is there any evidence that some vision experts are unnecessary? This question mostly arises from the consistently good probing performance for the image-to-text generation expert, hence I wonder if this can be removed with little to no performance degradation;
4. Were multiple runs executed for these experiments? If so, can the authors please report the standard deviation as well? This would be helpful to understand which performance gains are significant and which are not.
5. How are the sets of layers contributing to embedding distillation from different vision experts selected for the experiments with Phi3-4k-mini (Table 1)?

I am definitely open to increasing my score after a productive rebuttal.

**Ethical Concerns:**

["NO or VERY MINOR ethics concerns only"]

**Final Justification:**

The discussion with the authors was positive, but some of my primary concerns (primarily regarding the transfer of visual understanding capabilities, the significance of the reported outcomes, and the generalizability of design choices to other LLM architectures) remain partially unaddressed. I do, however, acknowledge that this paper has merits as well. I am therefore inclined to lean towards acceptance and keep my original "Borderline Accept" rating.

**Limitations:**

This work contains a Limitations section in the main body of the paper, which I think is sufficient.

**Paper Formatting Concerns:**

I do not see any formatting concerns.

**Quality:**

3

**Strengths And Weaknesses:**

**Strengths.**
- This paper tries to improve Visual Perception in Multimodal LLMs, which I believe is an importance aspect due to their well-known overreliance on text;
- Pairing the embedding distillation objective with the standard language modeling objective allows to maintain strong multimodal understanding alongside improving visual perception;
- I think Section 3 (Preliminary Analysis) provides cool insights that can be useful for future research (*i.e.* results about layer importance according to the visual task, as well as probes improving consistently with the addition of data);
- I overall think the experimental section is comprehensive; I particularly liked the "data-matched" Cambrian-1 experiment and the comparison against a RADIO-ViT-L based LLaVA;


**Weaknesses.**
- I wonder how much some of the downstream results may suffer from the chicken-and-egg dilemma when it comes to CV-Bench. All preliminary analyses conduct extensive experiments on all CV-Bench tasks, which inform all design choices of VisPerLM, and the main evaluation for vision-centric capabilities is conducted on CV-Bench as well. I think it is important for this paper to test downstream performance on other vision-centric benchmarks (*e.g.* BLINK [a]) to ensure this confounder disappears, and check whether the improved vision-perception abilities are stable and consistent;
- Reading the paper, it is a bit unintuitive that all tokens within LLM decoder blocks are used for probing, *i.e.*, since the goal of the resampler is to probe visual representations, I wonder whether only visual tokens alongside task tokens may be sufficient (or even better), compared to also using text tokens.

**References.**
[a] Fu, Xingyu, et al. "Blink: Multimodal large language models can see but not perceive." ECCV 2024.

---

> ### Author Rebuttal · Authors · 2025-07-28
>
> We thank the reviewer for their feedback and comments.
> We are glad that they recognize the importance of our effort to improve MLLMs at visual perception and that *"Section 3 (Preliminary Analysis) provides cool insights that can be useful for future research"*. We are encouraged that they found the experimental section *"comprehensive"* and *"particularly liked the "data-matched" Cambrian-1 experiment and the comparison against a RADIO-ViT-L based LLaVA"*.
>
> We respond to their questions and concerns below:
>
> ### **Pt #1: VisPerLM shows improvements on BLINK benchmark**
>
> > Reviewer's Comment: I wonder how much some of the downstream results may suffer from the chicken-and-egg dilemma when it comes to CV-Bench. All preliminary analyses conduct extensive experiments on all CV-Bench tasks, which inform all design choices of VisPerLM, and the main evaluation for vision-centric capabilities is conducted on CV-Bench as well. I think it is important for this paper to test downstream performance on other vision-centric benchmarks (e.g. BLINK [a]) to ensure this confounder disappears, and check whether the improved vision-perception abilities are stable and consistent;
> >
> > How much does the downstream performance of VisPer-LM suffer from informed decisions based on CV-Bench? Does improved visual perception persist when evaluating on other vision-centric benchmarks?
>
> Thank you for the suggestion to evaluate our method on the BLINK benchmark! We report results on visual perception tasks under the BLINK benchmark (`val` set) in the table below. We use CLIP-ConvNeXT-XXL and Llama3-8b as the base visual encoder and base LLM, respectively. We also report results using the data-matched Cambrian-1 RADIO-ViT-L based LLaVA-1.5 models for the reviewer's reference.
>
> | Method                | Spatial Relation | Relative Depth |  Avg  |
> |-----------------------|:----------------:|:--------------:|:-----:|
> | LLaVA-1.5             |       72.0       |      51.4      | 61.7  |
> | RADIO-L LLaVA-1.5     |       65.7       |    **52.4**    | 59.1  |
> | Cambrian-1            |       74.1       |      51.6      | 62.9  |
> | **VisPer-LM (ours)**  |     **75.5**     |      51.6      |**63.6**|
>
> We find that our VisPer-LM significantly outperforms all other baselines on average, especially on the `Spatial Relation` task, demonstrating the effectiveness of our approach.
>
> ### **Pt #2: Effect of system prompt and text tokens on the probing performance**
>
> > Reviewer's comment: Reading the paper, it is a bit unintuitive that all tokens within LLM decoder blocks are used for probing, i.e., since the goal of the resampler is to probe visual representations, I wonder whether only visual tokens alongside task tokens may be sufficient (or even better), compared to also using text tokens.
> >
> >Did the authors experiment with other probing setups (.e.g. omitting textual tokens as well as / or tokens from the system prompt)?
>
> We clarify that the goal of our probing setup is to evaluate the representation quality of all the tokens from the LLM as a whole and not just the tokens corresponding to the visual features. This is an important detail as prior works [32, 72] have shown the importance of system and text tokens during the LLM's token generation process. Therefore, in order to establish a correlation between the MLLM's representation quality and downstream performance, we believe it is critical to probe all tokens from a given sequence.
>
> Nonetheless, we agree this is an interesting experiment! Therefore, we share the average probe scores (across all layers) for our VisPer-LM model under different token input settings to the resampler below:
>
> | Input to Probe Resampler                  | Depth   |  Seg   |  Gen   |
> |------------------------------------------|:-------:|:------:|:------:|
> | &lt;img&gt; &#124; &lt;t&gt;                   | **0.613** | **0.564** | 0.744 |
> | &lt;img&gt; &#124; &lt;t&gt; &#124; &lt;text&gt;   | 0.608   | 0.562  | 0.748  |
> | &lt;sys&gt; &#124; &lt;img&gt; &#124; &lt;t&gt; &#124; &lt;txt&gt; | 0.603   | 0.563  | **0.764** |
>
> &lt;sys&gt;: system tokens; &lt;img&gt;: visual tokens; &lt;t&gt;: task-specific tokens; &lt;txt&gt;: text tokens
>
> As expected, dropping the system and text tokens, leads to the best depth and seg probing scores. Interestingly, using all tokens leads to the best gen probing performance which we attribute to the language aligned nature of the gen expert features (L#147 in the main text).
>
> However, note that we also conduct a similar experiment while training the model in appendix A Tab. III, i.e., ablating the tokens input to the embedding predictors for the auxiliary losses. We found that using all tokens as input to the embedding predictor leads to the best performance owing to the importance of the system and text tokens for LLM's reasoning.
>
> [32] Omri Kaduri, Shai Bagon, and Tali Dekel. What’s in the image? a deep-dive into the vision of vision language models. arXiv, 2024.
>
> [72] Guangxuan Xiao, Yuandong Tian, Beidi Chen, Song Han, and Mike Lewis. Efficient streaming language models with attention sinks. ICLR, 2024
>
> ### **Pt #3: RADIO based LLaVA**
>
> > Reviewer's Comment: About the RADIO-based LLaVA: did the authors train it themselves, or does it come from some previous work I am unaware of?
>
> We trained our own version of RADIO ViT-L LLaVA-1.5 based on Llama3-8b. We will clarify this in the main text. Thank you for pointing this out!
>
> ### **Pt #4: Removing the Gen expert**
>
> > Reviewer's Comment: Is there any evidence that some vision experts are unnecessary? This question mostly arises from the consistently good probing performance for the image-to-text generation expert, hence I wonder if this can be removed with little to no performance degradation;
>
> This is an interesting experiment! We provide ablations on the different combinations of auxiliary losses during training in appendix A (L#539-545).
> We share the analysis from the article here for the reviewer's reference and direct the reviewer to Tab. IV in the appendix for the numbers:
>
> *Our results reveal that the optimal performance is achieved when all three embedding losses: depth, seg, and gen, are used together. Interestingly, we observe that utilizing only depth or gen embedding losses still leads to notable performance improvements, whereas relying solely on seg embedding loss does not yield significant gains. This suggests that different types of target information contribute uniquely to the distillation process. Investigating how the distillation of one type of target information influences the effectiveness of others presents an intriguing direction for future research.*
>
> ### **Pt #5: Fixed seed for experiments**
>
> > Reviewer's Comment: Were multiple runs executed for these experiments? If so, can the authors please report the standard deviation as well? This would be helpful to understand which performance gains are significant and which are not.
>
> Due to resource constraints, we could not perform multiple runs. However, to ensure experimental validity and significance, we perform all our experiments with a fixed seed as also pointed out under point 7 in the checklist.
>
> ### **Pt #6: Embedding Distillation Layers for Phi3-4k-mini**
>
> > Reviewer's Comment: How are the sets of layers contributing to embedding distillation from different vision experts selected for the experiments with Phi3-4k-mini (Table 1)?
>
> Because Phi3-4k-mini has the same number of layers as Llama3-8b (32 layers), we use the same set of layers for both the LLMs based on the ablation results from Tab. 4 in the main text. We verified that Phi3-4k-mini based models showed similar probing trends as that of Llama3-8b based models while making this decision.

---

> > ### Comment · Reviewer_Khkk · 2025-08-02
> > **Thank you for the rebuttal! Let's discuss**
> >
> > Dear Authors,
> >
> > Thank you for the detailed rebuttal! Let me reply below.
> >
> > **CVBench-informed design and transfer to other vision-centric benchmarks.** Thank you for reporting experiments on the "Spatial Relationships" and "Relative Depth" subtasks of BLINK. It looks like these results somewhat confirm that the big performance gap reported by Tab.1 of the paper might be a by-product of making explicit design decisions of VisPerLM informed by CV-Bench priors. For example:
> > - for the "Depth" subtask of CV-Bench 3D, Tab.1 shows  $>+6pp$ improvement over the "data-matched Cambrian-1" model, while here there is no improvement for the "Relative Depth" subtask;
> > - similarly, the improvement on the "Relation" subtask of CV-Bench 2D is much higher than that of the related "Spatial Relationships" subtask of BLINK.
> >
> > Of course, one needs data to perform preliminary analyses and make design choices, I understand that. After looking at these results, however, I am further convinced that this paper would benefit from a more comprehensive evaluation of the vision-centric capabilities of the resulting models, especially considering that visual perception is at the core of this work. If I am not mistaken, BLINK has 14 subtasks, all tailored to visual perception. Could you please clarify why only the results on two of them were reported?
> >
> > Please correct my understanding if you think I am mistaken or if I am missing something!
> >
> > **Token selection for probing.** Thank you for sharing these results. I further checked the token ablation experiment in the appendix, and my doubts were cleared. Thank you.
> >
> > **RADIO-based LLaVA.** Thanks for sharing this detail. Could you please also share the pretraining and instruction-tuning configurations used for this in-house model?
> >
> > **Unneeded or redundant experts.** Apologies for missing this experiment in the appendix! It does indeed look like using only the depth or the gen expert leads to performance that is comparable to that of using all 3 of them. The patterns shown by the table, though, are a bit unintuitive, for example:
> > - Using either depth or gen experts improves over LLaVA-1.5, but the depth+gen combo performs worse not only than VisPerLM-gen or VisPerLM-depth, but even worse than the base model itself;
> > - The segmentation expert looks detrimental in all experiments (e.g., LLaVA > LLaVA-seg; LLaVA-depth > LLaVA-depth-seg; LLaVa-gen > LLaVA-gen-seg), and only brings improvement when used in the full depth+seg+gen combo.
> >
> > Do you have any explanation or hypothesis for this unpredictable behaviour? Or, if only one run was executed here as well, do you think these might be noisy observations?
> >
> > **Fixed seed vs standard deviation of multiple runs.** Thank you for clarifying this aspect. I totally understand that compute might be a limiting factor, but please also understand that from a reviewer's perspective, ensuring that reported results are somewhat significant is crucial. I would strongly suggest striving for multiple runs in any revision of the paper.
> >
> > **Layer choices for Phi3-4k-mini.** Did you perhaps experiment with models where the LLM decoder has a significantly different number of decoder blocks? At the current stage, applying the VisPerLM pipeline might require extensive trial-and-error to choose what to distill at which layer. This might be prohibitively expensive, and negatively impact the applicability of VisPerLM. Probably, this paper would benefit from some form of "transfer" of design choices between model architectures. What are your thoughts on this?

---

> > > ### Author Response · Authors · 2025-08-03
> > > **Thank you for your reply and thoughtful comments! We hope our clarifications below help**
> > >
> > > We thank the reviewer for their reply. We are glad that our rebuttal helped clear their doubts about our probing setup among other things. We appreciate their suggestion for pushing for multiple runs in future and will keep that in mind. We reply to their remaining concerns below:
> > >
> > > ### **BLINK and CV-Bench measure perception abilities differently**
> > >
> > > Thank you for raising this thoughtful point!
> > >
> > > We believe there are various factors to consider when comparing the performance gains of VisPer-LM over the baselines on CV-Bench and BLINK Benchmark:
> > >
> > > - **Our Target Perception Tasks:** We remind the reviewer that in our work, we refer to the spatial and depth reasoning aspects of visual perception as our target tasks (as mentioned on L#56-58). Therefore, we only shared results for the `Relative Depth` and `Spatial Relation` tasks under BLINK. As pointed out by the reviewer, BLINK has 14 tasks but not all are relevant to measuring the effectiveness of VisPer-LM.
> > >     - For example, 9 tasks require multi-image reasoning abilty which would require a special training recipe for meaningul evaluation.
> > >         - We direct the reviewer to Fig. 5 in the BLINK paper for more information on tha nature of these tasks: `Visual Correspondence`, `Multi-view reasoninhg`, `IQ Test`, `Semantic Correspondence`, `Jigsaw`, `Forensics Detection`, `Visual similarity`, and `Functional correspondence`.
> > >     - Additionally, tasks like `Relative Reflectance` and `Object Localization` also require special training data which is absent from the models' training mix and therefore, LLaVA like models actually perform worse than a random chance baseline, making the tasks unfit for our evaluation purposes.
> > >     - We understand the `Counting` task may be considered relevant, so we include those results in a table below where our VisPer-LM agains shows significant improvement over the the single encoder and Cambrian-1 baselines.
> > >
> > > | Task| LLaVA-1.5 | RADIO-L LLaVA-1.5 | Cambrian-1 | **VisPer-LM (ours)** |
> > > |---|:---:|:---:|:-:|:--:|
> > > | Counting | 43.3 |**49.2** |45.0|**49.2**|
> > >
> > > - **Nature of Evaluation Samples:** Although the task names (spatial reasoning and relative depth) are similar under CV-Bench and BLINK, the nature of samples under both benchmark are setup differently.
> > >     - For example, a depth task question in CV-Bench reads like: *"Which object is closer to the camera taking this photo, the lamp (highlighted by a red box) or the books (highlighted by a blue box)?"* while under BLINK, a depth task question would be like: *"Which point is closer?"* (with two points annotated on the image).
> > >         - The BLINK question is much harder for our model to answer because they have not been trained to perceive points on an image. Additionally, the number of samples for each task under CV-Bench are almost three times as that of under BLINK. Therefore, we believe it is unfair to expect for the model to show same percentage gains on both benchmarks.
> > >         - For example, our VisPer-LM shows an improvement of 4.2% on the counting task under BLINK over the "data-matched" Cambrian-1 but a 0.2% boost on CV-Bench. Although, it's not one of our target tasks, we believe it is a critical observation to consider while understanding the results.
> > >
> > > ### **RADIO-L LLaVA-1.5 Setup**
> > >
> > > The RADIO-L LLaVA-1.5 baseline uses the same training setup as all the other models trained with two stages (single/multi encoder baselines and our VisPer-LM). We mention these details on L#236-239 in the main text.
> > >
> > > ### **Role of Experts**
> > >
> > > We agree with the reviewer that the experiments from Tab. IV in the appendix do show some unpredictable findings. We did try running these experiments with different seeds but all our runs resulted in similar numbers with minimal variance (less than 0.2%). We must admit that as authors, we do not have a good theory about this behavior other than that multiple expert feature distillation has its own set of unanswered questions for future research including how each expert feature affects the distillation of other expert features. As mentioned in the appendix, we hope the community can help us better understand this in the future.
> > >
> > > ### **Experiments with LLMs with different number of decoder blocks**
> > >
> > > Thank you for the valuable suggestion! Based on our experiments, we believe it is optimal to choose 2 middle layers per expert for the embedding distillation loss based on the probing trends. Nonetheless, it is advisable to conduct probing experiments on an MLLM to identify the optimal "middle" layers. However, with the popularity of Llama3-8b and Phi3-4k-mini models as decoder LLMs, we feel the current version of our work is already useful to the community and we will work on including more LLM decoders in our future versions!

---

> > > > ### Comment · Reviewer_Khkk · 2025-08-05
> > > >
> > > > Dear Authors,
> > > >
> > > > Thank you for the response and for sharing your thoughts! Let me reply below:
> > > >
> > > > **Results on BLINK.** Thank you for clarifying why you think some of the subtasks in BLINK might not be suited for evaluating VisPerLM. I agree with multi-image samples requiring special treatment and, therefore, being unsuitable.
> > > >
> > > > Moving on, about the following statement:
> > > > >[...] a depth task question in CV-Bench reads like: "Which object is closer to the camera taking this photo, the lamp (highlighted by a red box) or the books (highlighted by a blue box)?" while under BLINK, a depth task question would be like: "Which point is closer?" (with two points annotated on the image). The BLINK question is much harder for our model to answer because they have not been trained to perceive points on an image.
> > > >
> > > > Although I see your point, I think this still suggests limited capabilities in abstracting and transferring the knowledge learned by VisPerLM. Judging from this example alone, the BLINK subtask looks exactly like the CV-Bench subtask, just in a different "visual format".
> > > >
> > > > While about the following:
> > > > >[...] the number of samples for each task under CV-Bench are almost three times as that of under BLINK. Therefore, we believe it is unfair to expect for the model to show same percentage gains on both benchmarks.
> > > >
> > > > Apologies, but I do not understand this point. Although it is of course better to evaluate on benchmarks with as many data points as possible, percentage points remove the dependency from the raw number of samples, don't they? Am I missing something here?
> > > >
> > > > At the same time, I think the results on the counting task are promising, and a nice result to show in the revision of the paper together with the results on `Spatial Relationships` and `Relative Depth`.
> > > >
> > > > **Radio-LLaVA v1.5 setup.** L#236-239 highlight the Pre-Training and Instruction-Tuning setup of VisPerLM, as they also mention method-specific details such as the special task tokens. I would suggest introducing a dedicated "Baselines" paragraph somewhere in the experimental section, with details about the Radio-LLaVA model and the "data-matched Cambrian-1" model to enhance clarity and organization.
> > > >
> > > > **Unpredictable results when combining experts.** I see, and I understand it might be difficult to come up with reasonable explanations.
> > > >
> > > > **Transfer of design choices to LLM decoders of varying sizes.** I agree with you that this work has enough value for the community by experimenting with LLaMA3-8B and Phi3-4K-Mini already, but I still think it'd be valuable to come up with design choices that are inherently more transferable without expensive trial-and-error.
> > > >
> > > >
> > > > All the best,
> > > > Reviewer `Khkk`

---

> ### Author Response · Authors · 2025-08-06
>
> We thank the reviewer for their reply and time! We appreciate their suggestions on adding more details for a dedicated `Baselines` section under implementation and will incorporate it in our revised version! We are also grateful for their valuable feedback on the importance of establishing a positive transfer of design choices between different LLM architectures and will spend time doing so!
>
> We provide clarification about the BLINK v/s CV-Bench point from our last comment below:
>
> > Reviewer's Comment: Although I see your point, I think this still suggests limited capabilities in abstracting and transferring the knowledge learned by VisPerLM. Judging from this example alone, the BLINK subtask looks exactly like the CV-Bench subtask, just in a different "visual format".
>
> We agree with the reviewer that the BLINK subtask is in a different visual format. However, we would like to point out that it is also in a different language format. We believe that because the LLM is responsible for reasoning over the given visual input and that the model has not seen such "which point is closer?" questions during training, the model is unable to understand the format well to show gains of the improved visual embedding. On the contrary, for CV-Bench, the questions contain object names which are easier for the model to reason over owing to the training samples from the visual genome and COCO based datasets. This observation may point to the over-dependency of MLLMs on the underlying LLMs for visual reasoning.
>
> Nonetheless, we understand the reviewer's point that VisPer-LM should be able to reason well over the BLINK's sample, however, the LLM-driven nature of MLLM makes it a bit hard for the model in our opinion, resulting in accuracy near chance behavior (i.e, 50% as the possible answers are only `yes` or `no` for Depth task under BLINK) for all models. We will look more into this though. We thank the reviewer for raising this point!
>
> > Reviewer's Comment: Apologies, but I do not understand this point. Although it is of course better to evaluate on benchmarks with as many data points as possible, percentage points remove the dependency from the raw number of samples, don't they? Am I missing something here?
>
> We apologize for our statement being unclear to the reviewer. We meant to share the argument about different number of samples in both benchmarks as a secondary factor and not the primary argument (primary being the nature of benchmarks). We agree that the dependency on % does remove the dependency from the raw number of samples. However, only to an extent. We believe the observations from approx 250 samples from the BLINK benchmark have a higher chance of being noisy than from about 700 samples under CV-Bench. Therefore, we believe the nature of questions is more at play here than the number of samples.
>
> > At the same time, I think the results on the counting task are promising, and a nice result to show in the revision of the paper together with the results on Spatial Relationships and Relative Depth.
>
> Definitely! We thank the reviewer for their continued suggestions and feedback. We will add these results to the revised version.
>
> We hope the above clarifications help and appreciate the reviewer's valuable feedback and interest in our work!

---

### Official Review · Reviewer_jTWX · 2025-07-01

**Clarity:** 2
**Significance:** 2
**Originality:** 2
**Rating:** 4
**Confidence:** 4

**Summary:**

Current  MLLMs is trained with natural language supervision, causing models to ignore the rich visual perception signals present in the data. This work finds that the quality of visual representations is positively correlated with the performance on downstream tasks. Therefore, the authors proposes to optimizes visual embedding prediction alongside next (text) token prediction during training. Specifically, it adds supervision from target visual information to the embeddings of selected LLM layers to improve visual embedding prediction. The proposed method, VisPer-LM, outperforms both single- and multi-encoder baseline methods on various benchmarks.

**Questions:**

1. During training, computing the three target probe features takes time. Compared to the multi-encoder approach, is the training time actually reduced? Specifically, where is the efficiency improvement reflected?
2. In the results shown in Figure 3, for the first and third rows, the performance of VisPer-LM is almost indistinguishable from the single encoder in the deeper LLM layers, and in the generation task, the single encoder even performs slightly better. This suggests that training with these three tasks does not significantly improve the quality of the visual embeddings. Therefore, I have doubts that the performance improvement on the benchmarks is actually due to the use of more data and an extended training stage, rather than an improvement in the quality of the three target visual embeddings.
3. Why does the multi-encoder have the highest visual embedding quality in Figure 3, but perform worse than VisPer-LM in Table 1? Doesn’t this contradict the paper’s claim that the quality of visual representations is positively correlated with downstream task performance?
4. As training progresses and the number of steps increases, does the quality of the visual embeddings and the model’s performance continue to improve accordingly?

**Ethical Concerns:**

["NO or VERY MINOR ethics concerns only"]

**Final Justification:**

The rebuttal addresses my concerns. I am now convinced that the performance improvements are primarily due to the enhanced visual embedding quality. I will update my scores to Borderline accept.

**Limitations:**

yes

**Quality:**

3

**Strengths And Weaknesses:**

Strengths
1. This paper proposes a method to enhance visual representations by distilling three target visual information, achieving a trade-off between efficiency and representation quality in both the single encoder and multi-encoder settings.
2. Extensive experiments demonstrate the effectiveness of the proposed approach.
3. Additional experiments analyze the choice of key parameters and the design of the model structure.

Weakness
1. The training time for each stage and the efficiency compared to the multi-encoder are not reported. The total loss includes three additional depth, segmentation, and generation tasks, respectively. Compared to the multi-encoder approach, is the training time actually reduced?
2. Since both increasing the training data and distilling the three target visual information are performed simultaneously, a further comparative explanation of the impact of these two operations is needed.
3. In Figure 3, it is confusing that the multi-encoder baseline has the best probing performance but achieves lower performance than VisPer-LM. Also, the probing performance of VisPer-LM that training with these three tasks does not significantly improve the quality of single encoder.

---

> ### Author Rebuttal · Authors · 2025-07-28
>
> We thank the reviewer for their feedback and comments.
> We are glad that they acknowledge the effectiveness of our approach supported by extensive experiments and analysis.
>
> We respond to their questions and concerns below:
>
> ### **Pt #1: Training Time and Inference Time Efficiency**
>
> > Reviewer's Comment: The training time for each stage and the efficiency compared to the multi-encoder are not reported. The total loss includes three additional depth, segmentation, and generation tasks, respectively. During training, computing the three target probe features takes time. Compared to the multi-encoder approach, is the training time actually reduced? Specifically, where is the efficiency improvement reflected?
>
> We would like to clarify that by performance-efficiency trade-off in the main text (L#48), we mainly refer to the inference time efficiency. Nonetheless, we report numbers for both inference and training time below to demonstrate the efficient nature of VisPer-LM compared to the multi-encoder baseline.
>
> **Inference time:** We demonstrate the superior inference throughput of VisPer-LM over the multi-encoder approach in Tab. VIII in appendix A. We report the numbers again below for the reviewer's reference:
>
> | inference                  | single-encoder      | multi-encoder      | **VisPer-LM (ours)**        |
> |---------------------------|---------------------|--------------------|----------------------|
> | throughput (samples/sec)  | 9.92 ± 0.03       | 5.32 ± 0.02       | 9.86 ± 0.01       |
>
> Our VisPer-LM has superior inference throughput compared to that of the multi-encoder baseline and similar throughput to that of the single-encoder baseline with much better performance. We record the throughput on a single NVIDIA 80G A100 GPU for a single forward pass on the CV-Bench evaluation set with a batch size of 1. We report the mean and standard deviation across 10 runs.
>
> **Training time:** We report stage-wise training time for the different approaches below for CLIP-ConvNeXT-XXL as our base visual encoder and Llama3-8b as the llm decoder on 8 A100 GPUs.
>
> | training                 | single-encoder      | multi-encoder      | **VisPer-LM (ours)**        |
> |---------------------------|---------------------|--------------------|----------------------|
> | PT stage   | 9 hours       | 16 hours       | 16 hours       |
> | IFT stage  | 15 hours       | 23 hours       | 15 hours       |
> | total training time (PT+IFT) | 24 hours       | 39 hours       | 31 hours       |
>
> As shown in the above table, our VisPerLM reduces the total training time by 8 hours as compared to the multi-encoder approach. Note that we only use the auxiliary losses during the PT stage, resulting in IFT training time comparable to that of the single encoder approach.
>
> ### **Pt #2: All baselines (single and multi encoder) use the same training data and steps as VisPerLM**
>
> > Reviewer's Comment: Since both increasing the training data and distilling the three target visual information are performed simultaneously, a further comparative explanation of the impact of these two operations is needed.
>
> We clarify that we use the same amount of training data and steps as the baselines for the numbers reported in Tab. 1 in the main text while demonstrating the superior performance of VisPerLM. Moreover, we show the impact of extra training data in Tab. 3 (with an extra VPT stage) in the main text, which also shows gains, showing the scalability of VisPerLM with more data. Therefore, our experiments already include separate experiments to demonstrate the inidividual effect of visual embedding distllation and increasing the amount of data.
>
> ### **Pt #3: The probing experiments explain the multi-encoder performance on the Target tasks**
>
> > Reviewer's Comment: In Figure 3, it is confusing that the multi-encoder baseline has the best probing performance but achieves lower performance than VisPer-LM. Also, the probing performance of VisPer-LM that training with these three tasks does not significantly improve the quality of single encoder.
> >
> > Why does the multi-encoder have the highest visual embedding quality in Figure 3, but perform worse than VisPer-LM in Table 1? Doesn’t this contradict the paper’s claim that the quality of visual representations is positively correlated with downstream task performance?
>
> That is a good question and observation! As explained on L#267-269 in the main text, the multi-encoder baselines are slightly better than VisPer-LM on the Depth task in CV-Bench owing to their superior depth representation quality, aligned with our claim that the *quality of visual representations is positively correlated with the downstream performance on the target tasks (Depth and Relation under CV-Bench)*. Note that we probe the LLaVA-1.5 (feat concat.) baseline in Fig. 3.
>
> The multi-encoder baseline performs worse on average than our VisPerLM in Tab. 1 as the former is unable to retain general reasoning ability (on MMStar) same as the single-encoder baseline, unlike VisPerLM, demonstrating the effectiveness of our approach.
>
> ### **Pt #4: Scalability with More training data**
>
> > Reviewer's Comment: As training progresses and the number of steps increases, does the quality of the visual embeddings and the model’s performance continue to improve accordingly?
>
> Yes, as shown in the second row of Fig. 2(a) of the main text, as we increase the amount of training data (and therefore, the number of steps), the probing score and downstream performance gradually increase as expected. We observe the same phenomenon in row 2 of Fig. 3 as explained on L#156-170.
>
> ### **Pt #5: Gen Probing Performance has low correlation with downstream performance**
>
> > Reviewer's Comment: In the results shown in Figure 3, for the first and third rows, the performance of VisPer-LM is almost indistinguishable from the single encoder in the deeper LLM layers, and in the generation task, the single encoder even performs slightly better. This suggests that training with these three tasks does not significantly improve the quality of the visual embeddings. Therefore, I have doubts that the performance improvement on the benchmarks is actually due to the use of more data and an extended training stage, rather than an improvement in the quality of the three target visual embeddings.
>
> As explained under **Pt #2**, we use the same amount of data for training our VisPerLM model as the single-encoder baseline. Therefore, the the auxliary losses are solely responsible for the observed performance and visual representation quality improvement.
>
> Moreover, in appendix D, we find a high positive correlation of 0.98 between the gen/seg probing scores and downstream performance on CV-Bench while only a correlation of 0.37 for the gen probing scores. Therefore, single encoder baseline having slightly better gen probing scores is not a surprise due to the nature of our target tasks (Depth and Relation under CV-Bench).
>
> As explained on L#143 in the main text, prior works [32, 33, 61] have shown that early-to-middle layers have the most significant effect on the reasoning performance in (M)LLMs, and our VisPerLM shows far superior (depth/seg) representation quality in those layers compared to the single encoder baseline, explaining its superior performance. Nonetheless, understanding how different layers contribute to an MLLM's reasoning abilities remains an interesting research direction for the future.
>
> [32] Omri Kaduri, Shai Bagon, and Tali Dekel. What’s in the image? a deep-dive into the vision of vision language models. arXiv, 2024.
>
> [33] Guy Kaplan, Matanel Oren, Yuval Reif, and Roy Schwartz. From tokens to words: On the inner lexicon of llms. arXiv, 2024.
>
> [61] Oscar Skean, Md Rifat Arefin, and Ravid Shwartz-Ziv. Does representation matter? exploring intermediate layers in large language models. In Workshop on Machine Learning and Compression, NeurIPS 2024.

---

> > ### Comment · Reviewer_jTWX · 2025-08-04
> >
> > Thank you for the clear and thorough rebuttal, which addresses my concerns. I am now convinced that the performance improvements are primarily due to the enhanced visual embedding quality. I will update my scores accordingly.

---

> > > ### Author Response · Authors · 2025-08-06
> > >
> > > We thank the reviewer for their reply and time! We are glad that our rebuttal helped to convince them about the effectiveness of our visual embedding distillation approach. We truly appreciate their willingness to update their scores!

---

### Official Review · Reviewer_KZsA · 2025-07-07

**Clarity:** 4
**Significance:** 2
**Originality:** 3
**Rating:** 4
**Confidence:** 5

**Summary:**

* This paper identifies that standard MLLM training with only a next-token prediction objective can result in models with underdeveloped visual perception.
* To address this, the authors propose VisPer-LM, a novel training approach that distills knowledge from multiple “expert” visual encoders (eg, for depth, segmentation) directly into the intermediate hidden layers of the LLM.
  * This is accomplished via an auxiliary embedding loss during the pre-training stage (and *only* this stage)
  * ⇒ This is a key advantage of this method: the expert encoders are only used during training, adding no latency at inference time.
* The authors first use a probing methodology to establish a correlation between internal visual representation quality and downstream performance, and then show that VisPer-LM improves performance on perception-heavy benchmarks compared to standard MLLM baselines.

**Questions:**

1. **Dimensionality of the "Generation" Probe Target**
   - In Section 4.1, you note that the target for the "generation" features (Etextgen) has only one token, unlike the 576 tokens for the depth and segmentation features
   - Could this significant difference in the prediction target's dimensionality explain the highly stable and distinct probing performance observed for the gen features in Figure 3?
2. **Using Multi-Token Generative Features for a Cleaner Comparison**
   - Following up on Q1… Other works have successfully extracted multi-token features from generative model
      - E.g., Cambrian-1 uses a 1024-token representation from Stable Diffusion  [[paper]](https://arxiv.org/abs/2406.16860)  [[diffusion_encoder.py]](https://github.com/cambrian-mllm/cambrian/blob/main/cambrian/model/multimodal_encoder/diffusion_encoder.py)
      - → from their code, it seems this is based on methods from "Emergent Correspondence from Image Diffusion" (DIFT)  [[paper]](https://arxiv.org/abs/2306.03881)  [[DIFT code]](https://github.com/Tsingularity/dift)
   - Have you considered using a similar multi-token generative feature extractor? This would create a much cleaner comparison against your multi-token depth and segmentation experts and help isolate the effect of *what* kind of knowledge is being distilled from the *dimensionality* of the distillation target.

**Ethical Concerns:**

["NO or VERY MINOR ethics concerns only"]

**Final Justification:**

Most of my concerns are addressed. This is a good paper worthy of acceptance.

**Limitations:**

yes

**Quality:**

4

**Strengths And Weaknesses:**

**Strengths:**

1. Introduces a novel and clever method for improving the visual perception of MLLMs. The core idea of distilling knowledge from expert encoders directly into the LLM's intermediate layers is well-motivated and insightful
2. Provides an excellent trade-off between performance and efficiency. It successfully infuses rich visual information into the model during training without incurring the inference-time latency of traditional multi-encoder approaches
3. Supported by a strong and novel probing analysis (Section 3\) that empirically grounds the motivation for the proposed method by correlating internal representation quality with downstream task performance
4. The paper is well-executed, with extensive ablations that explore the key design choices of the proposed method

**Weaknesses:**

1. The final performance gains on the main benchmarks, while consistent, are relatively modest (e.g., \~2.5% average improvement in Table 1\)
   - ⇒ While the methodological contribution is strong, the practical impact in terms of state-of-the-art performance is not dramatic
2. While the distillation method is effective for the targeted perception tasks, its reliance on a curated set of specialized "expert" encoders raises questions about its generality.
   - It's unclear if this approach is a scalable path forward for improving MLLMs broadly, or if it imposes too much specific structure, potentially at the expense of more general reasoning capabilities that might emerge from simpler, larger-scale training.

---

> ### Author Rebuttal · Authors · 2025-07-28
>
> We thank the reviewer for their feedback and comments.
> We are glad they found our idea *“well-motivated and insightful”* and our method *“novel and clever.”*
> We appreciate their recognition of our method's *“excellent trade-off between performance and efficiency”* and the *“empirically grounded motivation from a strong and novel probing analysis.”*
> We are also grateful for their positive remarks on the our work's execution and extensive ablations.
>
> We respond to their questions and concerns below:
>
> ### **Pt #1: Dimensionality of the "Generation" Probe Target**
>
> > Reviewer's Comment: In Section 4.1, you note that the target for the "generation" features (Etextgen) has only one token, unlike the 576 tokens for the depth and segmentation features.
> >
> > Could this significant difference in the prediction target's dimensionality explain the highly stable and distinct probing performance observed for the gen features in Figure 3?
>
> That is an interesting point and perspective! In the main text (L#147-148), we attribute the superior gen probing performance to the choice of $\textbf{E}^{\text{gen}}$ (from unCLIP-SD-2.1 [56]) which produces visual features that are already language-aligned owing to its training setup, making it easier for the probes to learn a mapping from the LLM representation (already in the language space).
>
> Therefore, we believe in this particular case, the nature of expert features plays the major role and not the number of tokens. Nonetheless, ablating different sets of expert encoders for probing and embedding distillation remains a promising direction for future work. We thank the reviewer for this intriguing question!
>
> [56] Aditya Ramesh, Prafulla Dhariwal, Alex Nichol, Casey Chu, and Mark Chen. Hierarchical text-conditional image generation with clip latents. arXiv, 2022.
>
> ### **Pt #2: Using Multi-Token Generative Target Features**
>
> > Reviewer's Comment: Following up on Q1… Other works have successfully extracted multi-token features from generative model
> >
> > E.g., Cambrian-1 uses a 1024-token representation from Stable Diffusion
> > from their code, it seems this is based on methods from "Emergent Correspondence from Image Diffusion".
> >
> > Have you considered using a similar multi-token generative feature extractor? This would create a much cleaner comparison against your multi-token depth and segmentation experts and help isolate the effect of what kind of knowledge is being distilled from the dimensionality of the distillation target.
>
> Thank you for your suggestion and sharing the exact reference! We share results with the usage of 1024-token outputs from the SD-2.1 as the gen expert features in the table below. We observe that the features from unCLIP-SD-2.1 (used in the main text) performs much better than using the 1024 token representation from SD-2.1 (VAE+UNet). Therefore, we believe that the information captured by the feature embeddings plays a more critical role than their dimensionality and a strong generative prior may even harm the MLLM's understanding performance in some cases, therefore it is important to choose the experts wisely. Nonetheless, future research into comparison of different experts is an intriguing direction.
>
> | gen expert features             | Depth | Relation | Count | Distance | MMStar | Avg   |
> |--------------------------------|:-----:|:--------:|:-----:|:--------:|:------:|:-----:|
> | single token from unCLIP-SD-2.1 | 69.4 |  74.2    | 51.3  |   54.3   |  39.5  | **57.7** |
> | 1024 tokens from SD-2.1 (VAE+UNet) | 61.2  |  66.8    | 47.6  |   55.7   |  35.8  | 53.4 |
>
> ### **Pt #3: Impact of VisPerLM**
>
> > Reviewer's Comment: The final performance gains on the main benchmarks, while consistent, are relatively modest (e.g., ~2.5% average improvement in Table 1)
> >
> > ⇒ While the methodological contribution is strong, the practical impact in terms of state-of-the-art performance is not dramatic
>
> We appreciate the reviewer’s recognition of our methodological contribution's strength. Enhancing MLLMs in visual perception and reasoning remains an active area of research, with several recent works—such as Cambrian-1 [66], which leverages multiple encoders to address the limitations of single-encoder MLLMs, and AM-RADIO [57], which distills knowledge from multiple teacher encoders into a single, enhanced encoder.
>
> We compare our VisPerLM against these approaches in Table 1 of the main text, demonstrating that our method outperforms both under identical data settings, highlighting the effectiveness of our proposed approach.
>
> We believe a 2.5% improvement on average is not modest in an academic setting where we do not use any extra/different data to train the MLLM compared to the baselines. However, we acknowledge the reviewer’s concern regarding the practical impact in terms of state-of-the-art performance. As noted in our limitations, we were unable to conduct experiments at larger scale due to resource constraints. Nonetheless, we believe the current results provide strong empirical evidence supporting a new paradigm for improving MLLMs. We also hope our probing framework serves as a valuable tool for understanding the underlying factors driving performance gains for the community.
>
> [66] Shengbang Tong, Ellis Brown, Penghao Wu, Sanghyun Woo, Manoj Middepogu, Sai Charitha Akula, Jihan Yang, Shusheng Yang, Adithya Iyer, Xichen Pan, Austin Wang, Rob Fergus, Yann LeCun, and Saining Xie. Cambrian-1: A fully open, vision-centric exploration of multimodal llms, NeurIPS 2024.
>
> [57] Mike Ranzinger, Greg Heinrich, Jan Kautz, and Pavlo Molchanov. Am-radio: Agglomerative vision foundation model reduce all domains into one. In CVPR, 2024
>
> ### **Pt #4: VisPerLM does not harm general reasoning**
>
> > Reviewer's Comment: While the distillation method is effective for the targeted perception tasks, its reliance on a curated set of specialized "expert" encoders raises questions about its generality.
> >
> > It's unclear if this approach is a scalable path forward for improving MLLMs broadly, or if it imposes too much specific structure, potentially at the expense of more general reasoning capabilities that might emerge from simpler, larger-scale training.
>
> We remind the reviewer that VisPerLM's core idea is to improve an MLLM at target perception tasks without any loss of its general reasoning abilities. For example, our approach provides a promising direction to improve current video understanding models at tasks like motion understanding. We leave the exploration of our method with large scale data and more target tasks to future work as noted under limitations.

---

> > ### Comment · Reviewer_KZsA · 2025-08-06
> >
> > Thank you for the reply. Most of my concerns are addressed.

---

> > > ### Author Response · Authors · 2025-08-06
> > >
> > > We thank the reviewer for their reply and time! We are glad that our rebuttal helped address their concerns. We are happy to respond to any remaining concerns, should they arise.

---

### Comment · Area_Chair_NpC2 · 2025-08-01
**Author-Reviewer Discussions (July 31 - Aug 6)**

Dear reviewers,

Authors have provided their rebuttals and now we are in Author-Reviewer Discussions (July 31 - Aug 6) period. Please read authors' rebuttal and give comments. Thanks!

---

### Decision · Program_Chairs · 2025-09-17

**Decision:**

Accept (poster)

**Comment:**

In this paper, authors proposed VisPer-LM that infuses visual perception knowledge from expert vision encoders into the LLM's hidden representations. They formulated the objective during the pretraining stage in MLLMs as a coupled optimization of predictive visual embeddings and next token prediction. They showed that VisPer-LM outperformed the single and multi-encoder baselines.

This paper got 4 borderline accept ratings.

The strength of this paper suggested by reviewers are:
1) proposed method is novel, clever, well-motivated and insightful. (Reviewer KZsA)
2) excellent trade-off between performance and efficiency. (Reviewer KZsA, jTWX)
3) well-executed, extensive ablations, extensive experiments. (Reviewer KZsA, jTWX, Khkk, JZQJ)
4) solves an important problem. (Reviewer Khkk, JZQJ)
5) provides cool insights (Reviewer Khkk)
6) paper is well written (Reviewer Khkk)

The weaknesses are:
1) final performance gains on the main benchmarks, while consistent, are relatively modest (Reviewer KZsA)
2) questions about its generality. (Reviewer KZsA)
3) training time for each stage is not reported. (Reviewer KZsA)
4) some additional analysis or experiments needed (Reviewer Khkk, JZQJ)

After rebuttal, Reviewer KZsA, jTWX, JZQJ thought their concerns are addressed by authors and gave borderline accept rating. Reviewer Khkk suggested that some of their concerns for example "primarily regarding the transfer of visual understanding capabilities, the significance of the reported outcomes, and the generalizability of design choices to other LLM architectures" remain partially unaddressed. but overall positive about this paper and gave borderline accept rating.

Given these AC decided to accept this paper.